



# A One-Dimensional Model of Turbulent Flow Through 'Urban' Canopies: Updates Based on Large-Eddy Simulation

Negin Nazarian [1], E. Scott Krayenhoff [2], and Alberto Martilli [3]

[1]Faculty of Built Environment, University of New South Wales, Sydney, Australia
[2]School of Environmental Sciences, University of Guelph, Guelph, Canada
[3]Center for Research on Energy, Technology and Environment (CIEMAT), Madrid, Spain

**Abstract.**

In mesoscale climate models, urban canopy flow is typically parameterized in terms of the horizontally-averaged (1-D) flow and scalar transport, and these parameterizations can be informed by Computational Fluid Dynamics (CFD) simulations of the urban climate at microscale. Reynolds Averaged Navier-Stokes Simulation (RANS) models have been previously employed to

derive vertical profiles of turbulent length scale and drag coefficient for such parameterization. However, there is substantial evidence that RANS models fall short in accurately representing turbulent flow fields in the urban roughness sublayer. When compared with more accurate flow modeling such as Large Eddy Simulations (LES), we observed that vertical profiles of turbulent kinetic energy and associated turbulent length scales obtained from RANS models are substantially smaller specifically in the urban canopy. Accordingly, using LES results, we revisited the urban canopy parameterizations employed in the one-

dimensional model of turbulent flow through urban areas, and updated the parameterization of turbulent length scale and drag coefficient. Additionally, we included the parameterization of the dispersive stress, previously neglected in the 1-D column model. For this objective, the PArallelized Large-Eddy Simulation Model (PALM) is used and series of simulations in an idealized urban configuration with aligned and staggered arrays are considered. The plan area density ($\lambda_p$) is varied from 0.0625 to 0.44 to span a wide range of urban density (from sparsely developed to compact midrise neighborhoods, respectively). In

order to ensure the accuracy of the simulation results, we rigorously evaluated the PALM results by comparing the vertical profiles of turbulent kinetic energy and Reynolds stresses with wind tunnel measurements, as well as other available LES and DNS studies. After implementing the updated drag coefficients and turbulent length scales in the 1-D model of urban canopy flow, we evaluated the results by a) testing the 1-D model against the original LES results, and demonstrating the differences in predictions between new (derived from LES) and old (derived from RANS) versions of the 1-D model, and b) testing the

1-D model against LES results for a test-case with realistic geometries. Results suggest a more accurate prediction of vertical turbulent exchange in urban canopies, which can consequently lead to an improved prediction of urban heat and pollutant dispersion at the mesoscale.





# 1 Introduction

Mesoscale meteorology is of particular interest for urban climate analysis: many weather phenomena that directly impact human activities occur at this scale; and the effects of urban roughness, heat, pollutant, and moisture on the atmospheric boundary layer (characterized as Urban Boundary Layer) have important mesoscale implications. Accordingly, mesoscale

modeling is a powerful tool for the analysis of urban climate, and further prediction and management of urban heat and pollution.

In mesoscale models, urban climate variables on timescales of hours to days depend on multiple spatial scales from the street-scale through to synoptic scales. Given contemporary computational resources, however, it is not feasible to explicitly resolves building shapes (O(1-100m)), and at the same time span a domain large enough to assess mesoscale impacts on the

UBL (O(10-100km)). Therefore, mesoscale models must parameterize the subgrid scale exchanges of momentum, pollutant, moisture and heat across the Urban Canopy Layer (UCL) and Urban Boundary Layer (UBL) interface (Fig. 1).

These 'subgrid'-scale urban processes may be classified as hydrodynamic (flow) or thermal (e.g., radiation, convection, conduction). In case of the former (focus of this study), the flow near the surface has been treated with approaches of varying complexity. The simplest and oldest is the bulk transfer approach, with Monin-Obukhov similarity theory (Monin and Obukhov,

1954) to account for varying atmospheric stability. However, this approach accounts only for surface-atmosphere exchange and the effects on the overlying atmosphere. Canopies (e.g., forests, urban neighbourhoods) result in a new atmospheric layer of importance: the roughness sublayer (Rotach, 1993), or its subset, the canopy layer (Oke, 1976). It is the flow in this layer that directly impacts the wind, air temperature, pollutant levels to which urban dwellers are exposed.

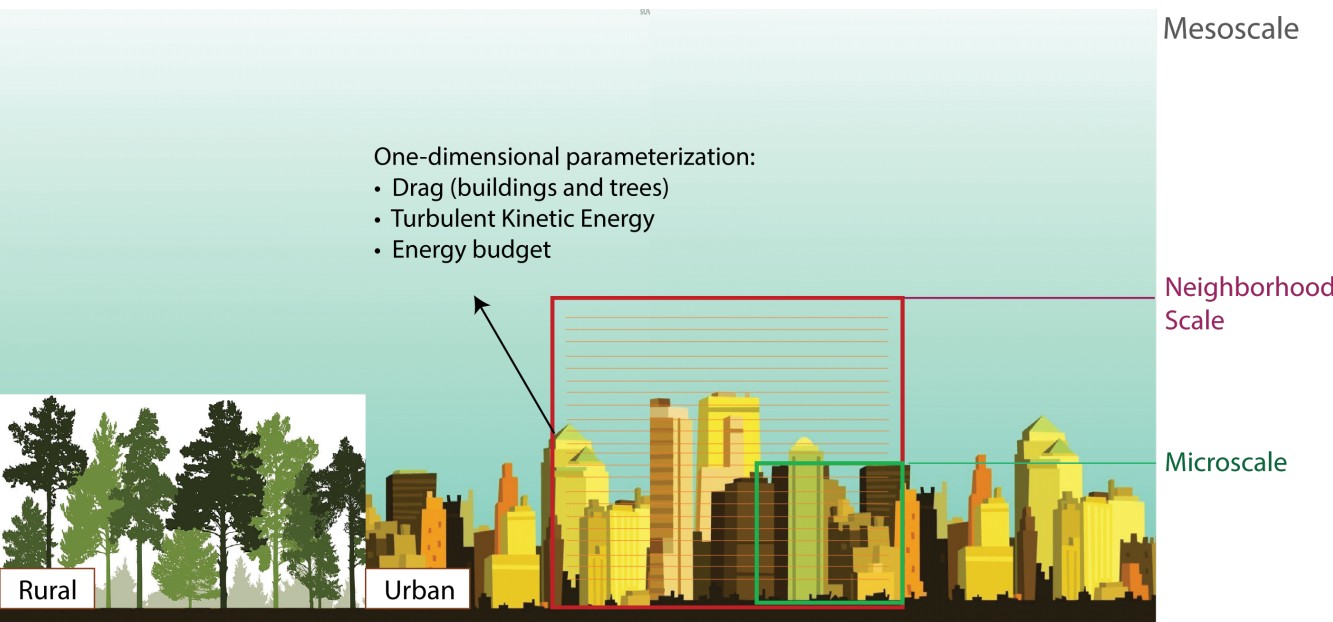

**Figure 1.** Schematic of Urban Canopy Parameterization (UCP) in multi-layer column models at mesoscale.





In the past two decades, urban canopy models (UCMs) have been developed to approximate the flow and thermal exchanges within and above neighborhoods and to couple with mesoscale models. Single-layer UCMs (Masson, 2000; Kusaka et al., 2001; Kanda et al., 2005; Bueno et al., 2013) have only one layer within the canopy, and focus on the overall exchange of heat, momentum, and moisture with the overlying atmospheric model. Moreover, they typically parameterize exchange

of momentum using MOST and use simple empirical relations to diagnose canopy wind speed. Multi-layer UCMs (Martilli et al., 2002) have several layers within the canopy (Fig. 1), and permit a more process-based treatment of canopy physics. However, they are computationally expensive as they employ prognostic equations for both momentum and turbulent kinetic energy (TKE) solved with 'urban canopy parameterization' or UCP (Martilli et al., 2002; Dupont et al., 2004; Santiago and Martilli, 2010). For instance, Santiago and Martilli (2010) present a one-dimensional (column) model of vertical exchange

of momentum and turbulent kinetic energy based on the $k - l$ turbulence closure scheme (1.5 order). This model employs horizontally-averaged microscale Computational Fluid Dynamics (CFD) simulations based on Reynolds-Averaged Navier-Stokes (RANS) to determine required input parameters to the column model (drag coefficients and turbulent length scales as a function of height), and is designed to predict the hydrodynamic component of multi-layer UCMs. Similarly, Simón-Moral et al. (2014) employed CFD simulations of idealized urban configurations, and updated the parameterization of drag

and turbulent length scale based on the horizontal heterogeneity cased by the variation of streanwise and spanwise streets. Krayenhoff et al. (2015) further extended the column model to include the effects of tree foliage on mean wind and turbulent kinetic energy in urban canopies. Subsequently, Krayenhoff et al. (2019) added temperature, humidity and buoyancy effects to the Krayenhoff et al. (2015) flow model (Santiago et al., 2014) and combined it with previously developed models on radiation (Krayenhoff et al., 2014) and thermal (Martilli et al., 2002) balance for a comprehensive representation of trees at the street

level.

The combined multi-layer urban canopy model, called BEP-Tree (Krayenhoff et al., 2019), is the first multi-layer (column) model of urban flow and energy exchange at the neighbourhood scale that includes the radiation and dynamic effects of trees in the street canyon. However, when compared with detailed spatially-sampled measurements over a 2 km by 2 km area in the Sunset neighborhood in Vancouver, Canada (Crawford and Christen, 2014), results indicated that the model strongly

overestimated daytime air temperature. Krayenhoff (2014) concluded that the underestimation of vertical exchange of heat is what results in a significantly higher canopy air temperature calculated. Additionally, the study reported that large differences persist with or without trees, and for several days of simulation; therefore, the underestimation can not be attributed to the parameterization of trees or anomalies in the observations. Recent work by Krayenhoff et al. (2019) demonstrates that larger turbulent length scales (based on the current LES analysis) markedly improve pedestrian level air temperature predictions

compared to measurements.

In this study, we aim to investigate the factors contributing to the underestimation of vertical exchange of heat and momentum in the multi-layer column model. We speculate that the following factors may be responsible:

- **RANS simulations as the basis for 1-D parameterization.** Given the simplified assumption of the turbulent flow in the RANS models, it is likely that the turbulent length scales derived from the RANS-CFD model are a culprit.

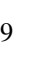




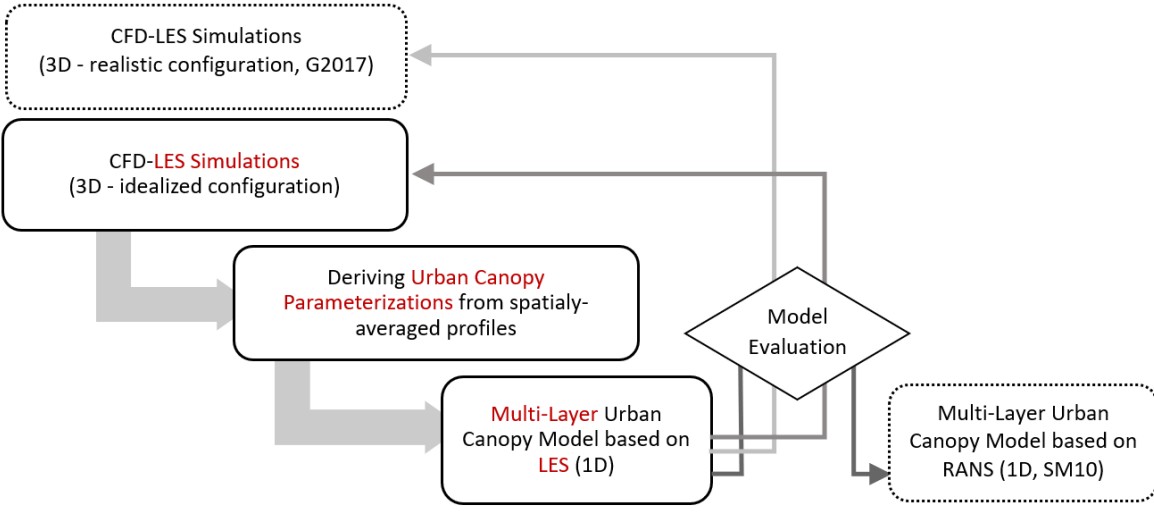

**Figure 2.** Flow chart of the present study. 3D LES simulations are performed and vigorously tested in an idealized configuration of buildings. Then, using the spatially-averaged profiles, Urban Canopy Parameterizations of the multi-layer (column) model are revisited. The updated multi-layer (1-D) model is then evaluated against the UCPs in Santiago and Martilli (2010), the 3D simulation results, as well as LES results in a realistic configuration by Giometto et al. (2017).

- **Contribution of dispersive stress.** The dispersive stress has been neglected in the parameterization and formulation of multi-layer model, though it has been shown (Coceal et al., 2006) that it contributes to the total turbulent flux at urban canopy level, specifically for higher urban densities.

- **Idealized versus realistic configurations.** So far, the parameterizations are derived for the simplified 'urban' arrays with uniform height, while mesoscale models aim to represent the impact of real urban neighborhoods.

- **Thermal effects.** It is posible that the simplistic representation of thermal effects on vertical turbulent heat transport further contributes to the underestimation of turbulent exchange.

Considering that a) the underestimation of vertical exchange of momentum is also seen in neutral cases, and b) the height variability in the Sunset neighborhood in Vancouver, which was used in Krayenhoff et al. (2019) for model evaluation, is relatively small, we focus on the first two factors in this analysis. Accordingly, for a more robust assessment of the urban canopy parameterization in the column model (focusing on the turbulent length scales, in particular), we employ a Large-Eddy Simulation (LES) model for a more accurate representation of turbulent flow (Xie and Castro, 2009; Salim et al., 2011; Gousseau et al., 2011; Nazarian et al., 2018a, b) and aim to include the contribution of dispersive stress.

Fig. 2 summarizes the structure of the present study. Sect. 2 describes the methodology to achieve the objectives in the 1-D multi-layer urban canopy model and in Sect. 2.2.2, the LES model and set-up are rigorously-tested to ensure fidelity of the results. Subsequently, drag coefficients and turbulent length scales are derived from LES of idealized arrays of cubes at





varying densities for neutral conditions (Sects. 3.3 and 3.2). Finally, the column model with updated parameters is evaluated against horizontally-averaged LES results in both idealized and realistic configurations (Sect. 3.4). Sect. 4 further summarizes the findings of this study and map out future developments of the multi-layer model.

## 2 Numerical Methods

### 5 2.1 One-dimensional Column $k - l$ Model

The momentum equation in mesoscale models undergoes two averaging processes (Martilli and Santiago, 2007; Santiago and Martilli, 2010). First, the Reynolds decomposition is applied to the momentum equation such that the mean flow quantities are separated from fluctuating turbulent parameters (time- or ensemble-averaging, $u = \overline{u} + u'$). Second, quantities are spatially averaged over volumes that can be compared to a grid cell of a mesoscale model (horizontal-averaging, $\overline{u} = \langle \overline{u} \rangle + \tilde{u}$). Addi-

tionally, assuming 1) horizontal homogeneity (and hence, zero mean vertical velocity due to the assumed incompressibility), 2) negligible Coriolis effect, and 3) negligible buoyancy effects, the equation for the horizontal momentum is presented as follows:

$$\frac{\partial \rho \langle \overline{u} \rangle}{\partial t} = -\frac{\partial \rho \langle \overline{u'w'} \rangle}{\partial z} - \frac{\partial \rho \langle \tilde{u}\tilde{w} \rangle}{\partial z} - \left\langle \frac{\partial \overline{P}}{\partial x_i} \right\rangle + \nu \langle \nabla^2 \tilde{u} \rangle, \tag{1}$$

where $u$ and $w$ are the streamwise and vertical velocity components, $P$ is the pressure, and $\rho$ is the air density (assumed to

be constant here). In this equation and onward, $< \psi >$ and $\overline{\psi}$ denote the spatial and time average of parameter $\psi$, respectively, and $\psi'$ and $\tilde{\psi}$ are the departure of the instantaneous parameter $\psi$ from the time or ensemble mean, and the deviation of the mean quantity $\overline{\psi}$ from its spatial average, respectively (i.e. $\tilde{\psi} = \overline{\psi} - \langle \overline{\psi} \rangle$ and $\psi' = \psi - \overline{\psi}$). More information on the averaging techniques can be found in Martilli and Santiago (2007).

Accordingly, the first term in the right-hand side (RHS) of Eq. 1 is the spatial average of the time-averaged turbulent fluxes,

while the second term is the dispersive stress (Raupach and Shaw, 1982; Martilli and Santiago, 2007), which accounts for the transport due to time-averaged structures smaller than the size of the averaging volume. Additionally, the third and fourth terms indicate the spatially-averaged acceleration due to the pressure gradient, as well as the spatial average of dispersive viscous dissipation (viscous drag), respectively.

To parameterize the contribution of the spatially-averaged turbulent momentum flux (first RHS term in Eq. 1), a K-theory

approach is used,

$$\langle \overline{u'w'} \rangle = -K_m \frac{\partial \langle \overline{u} \rangle}{\partial z}. \tag{2}$$

where $K_m$ is the diffusion coefficient for momentum using a $k - l$ closure (Martilli et al., 2002) as

$$K_m = C_k l_k \langle \overline{k} \rangle^{1/2} \tag{3}$$

where $C_k$ is a model constant for momentum, $l_k$ is a length scale, and $k$ is the turbulent kinetic energy (TKE). $C_k l_k$ is

parameterized in the column model based on the CFD results (further detailed in Sect. 3.3).





To calculate the spatially-averaged TKE in Eq. 3, a prognostic equation is then solved where the same assumptions as Eq. 1 are made. The resulting equation is

$$\frac{\partial \langle \overline{k} \rangle}{\partial t} = -\langle \overline{u_i' u_j'} \rangle \frac{\partial \langle \overline{u_i} \rangle}{\partial x_j} - \frac{\partial \langle \overline{k' w'} \rangle}{\partial z} - \frac{\partial \langle \tilde{k}\tilde{w} \rangle}{\partial z} - \left\langle \widetilde{u_i' u_j'} \frac{\partial \tilde{u}_i}{\partial x_j} \right\rangle - \frac{1}{\rho} \frac{\partial \langle \overline{p' u_i'} \rangle}{\partial x_i} - \langle \overline{\varepsilon} \rangle. \tag{4}$$

By a) parameterizing the shear production ($-\langle \overline{u_i' u_j'} \rangle \frac{\partial \langle \overline{u_i} \rangle}{\partial x_j}$) and turbulent transport terms ($-\frac{\partial \langle \overline{k' w'} \rangle}{\partial z}$) with K-theory (Eq. 3) and

b) assuming $K_m$ to be same for TKE and momentum (i.e. $K_m = -\frac{\langle \overline{u' w'} \rangle}{\partial \langle \overline{u} \rangle / \partial z} = -\frac{\langle \overline{k' w'} \rangle}{\partial \langle \overline{k} \rangle / \partial z}$), the TKE equation in the 1-D model is described as

$$\frac{\partial \langle \overline{k} \rangle}{\partial t} = K_m \left[ \left( \frac{\partial \langle \overline{u} \rangle}{\partial z} \right)^2 + \left( \frac{\partial \langle \overline{v} \rangle}{\partial z} \right)^2 \right] + \frac{\partial}{\partial z} \left( K_m \frac{\partial \langle \overline{k} \rangle}{\partial z} \right) - \frac{\partial \langle \tilde{k}\tilde{w} \rangle}{\partial z} - \left\langle \widetilde{u_i' u_j'} \frac{\partial \tilde{u}_i}{\partial x_j} \right\rangle + D_k - \langle \overline{\varepsilon} \rangle, \tag{5}$$

where $D_k$ is the source of $\langle \overline{k} \rangle$ generated through the interaction with the buildings and the air flow, and $\langle \overline{\varepsilon} \rangle$ is the viscous dissipation rate computed as

$$\langle \overline{\varepsilon} \rangle = C_\varepsilon \frac{\langle \overline{k} \rangle^{3/2}}{l_\varepsilon}. \tag{6}$$

$C_\varepsilon$ and $l_\varepsilon$ here are the model constant and the length scale of dissipation, respectively. In Santiago and Martilli (2010), $l_\varepsilon$ in Eq. 6 is derived from the CFD-modelled turbulent kinetic energy and dissipation (Eq. 6) and using the RANS model constant for turbulent viscosity ($C_\mu$), the turbulent length scale ($l_k$) is calculate as

$$C_k l_k = C_\mu \frac{l_\varepsilon}{C_\varepsilon}. \tag{7}$$

Accordingly, to solve prognostic Eqs. 1 and 5, two main parameterizations should be provided. First, the turbulent length scales (and consequently the dissipation length scales) are parameterized based on the CFD results of $\langle \overline{k} \rangle$, $\langle \overline{u' w'} \rangle$, and $\frac{\partial \langle \overline{u} \rangle}{\partial z}$ at different heights in the UCL (Eqs. 2 and 3). Second, the drag term due to buildings is parameterized as follows. In the momentum equation (Eq. 1), the drag force is introduced as a sink of momentum, given that buildings are not explicitly resolved and the averaging air volume is not connected (i.e. containing porosities representing the volume of the buildings).

Accordingly, the drag at height z is parameterized (Santiago and Martilli, 2010),

$$\left\langle \frac{\partial \overline{P}}{\partial x} \right\rangle \bigg|_z = D = S(z) C_d \langle \overline{u(z)} \rangle |\langle \overline{u(z)} \rangle|, \tag{8}$$

where $S(z)$ is sectional building area density ($\mathrm{m}^2$ of area facing the wind per $\mathrm{m}^3$ of outdoor air volume), $\langle \overline{u(z)} \rangle$ is the spatially-averaged mean wind speed, and $C_d$ is the sectional drag coefficient for buildings.

Additionally, analogous to the momentum equation, the source term of TKE due to the conversion of mean kinetic energy

into turbulent kinetic energy by the presence of buildings is parameterized as,

$$-\left\langle \widetilde{u_i' u_j'} \frac{\partial \tilde{u}_i}{\partial x_j} \right\rangle + D_k = S(z) C_d |\langle \overline{u(z)} \rangle|^3. \tag{9}$$

Similarly, the parameterization of drag induced by tree foliage and the interaction with the buildings can be considered, detailed in Krayenhoff et al. (2015). Using the LES results, we revisit the parameterization of length scales and drag coefficient induced by buildings, and discuss the consideration of dispersive stresses in Sects. 3.3 and 3.2.





## 2.2 Three-dimensional Large-Eddy Simulation Model

The LES results are used as a superior method to RANS models for evaluating turbulence characteristics and dispersion behavior in urban canopies (Xie and Castro, 2006; Salim et al., 2011; Nazarian and Kleissl, 2016). A PArallelized Large-Eddy Simulation Model (Raasch and Schröter, 2001; Letzel et al., 2008; Maronga et al., 2015) is employed here, which

solves the following: filtered incompressible Boussinesq equations, the first law of thermodynamics, passive scalar equation, and the equation for subgrid-scale (SGS) turbulent kinetic energy. The subgrid-scale fluxes are parameterized using the 1.5-order Deardorff flux–gradient relationships (Deardorff, 1980) which uses the SGS-TKE equation to calculate eddy viscosity. The Temperton algorithm for the fast Fourier transform (FFT) is also used to solve the Poisson equation for the perturbation pressure. A more detailed description of PALM can be found in Maronga et al. (2015).

### 2.2.1 Setup of LES Simulations

A series of neutral simulations is considered for idealized urban-like configurations with aligned (Fig. 3-a) and staggered (Fig. 3-b) arrays of identical cubes. The plan area density ($\lambda_p = A_p/A_T$) is varied from 0.0625 to 0.44 for both configurations to span a wide range of urban density (from sparsely developed to compact midrise neighborhoods, respectively), where $A_p$ and $A_T$ are the plan area and total area of roughness elements, respectively. Similar to Santiago and Martilli (2010), the obstacles

are cubes, such that $\lambda_p = \lambda_f$. Total height in the simulations domain is $7.4H$, where $H$ is the building height ($16\,\mathrm{m}$), and the wind direction is in the $x$ direction and perpendicular to the array (Fig. 3). The simulations are performed over arrays of $5\times3$ and $6\times3$ (Nx by Ny) buildings for the aligned and staggered configurations, respectively. In all simulations, the canyons height is resolved by 32 grids (representative of grid resolution of $0.03H$), and uniform grid resolution is used in $x$ and $y$ directions. In the vertical direction ($z$), a uniform grid is used up to $4H$ and grid spacing is gradually increased thereafter. Periodic boundary

conditions are employed in horizontal directions ($x$ and $y$ axes) to simulate an infinite array. The specifications of the geometric configuration, domain height, and grid resolution are motivated by detailed sensitivity analyses in Yaghoobian et al. (2014) and Nazarian et al. (2018a) to ensure that the large eddies influencing the canopy flow are resolved. Additionally, Sect. 2.2.2 further discusses the validity of simulation setups for the parameterization of canopy flow.

The flow is driven by a pressure gradient of magnitude $\tau = \rho u_\tau^2/H_T$, where $u_\tau$ is the total wall friction velocity, and $H_T$

is the total domain height ($7.4H$). The corresponding $u_\tau$ is $\approx 0.2\,\mathrm{m\,s^{-1}}$ which results in $Re_H = UH/\nu \approx 10^6$. For the top boundary condition for momentum, a zero-gradient (free-slip) boundary condition that enforces a parallel flow is used.

### 2.2.2 Model Evaluation: Validity of the Simulation Setups

The PALM model is widely used and has been validated against various experimental measurements (Maronga et al., 2015; Park et al., 2012; Yaghoobian et al., 2014; Nazarian et al., 2018b). However, since the parameterization of the multi-layer model

requires a high accuracy of results for the turbulent flow characteristics, we extended our analysis to evaluate the validity of the simulation setups for this study and further compare the results with the RANS model (Sect. 2.2.3). First, we compared the profiles of turbulent kinetic energy and Reynolds stresses with wind tunnel experiment data as well as other available LES



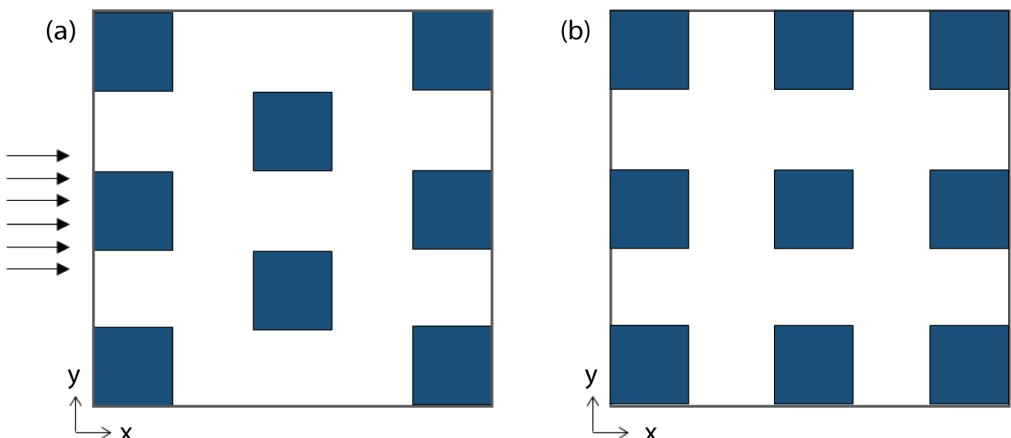

**Figure 3.** Plan view of configurations used for LES analyses, representing "staggered" (a) and "aligned" (b) arrays of buildings. Note that computational domains consist of 5×3 and 6×3 (Nx by Ny) buildings for the aligned and staggered configurations, respectively, and only a subsection of this domain is shown here.

studies. Our velocity and Reynolds stress showed good agreement when compared with the LES results of Kanda et al. (2004) (not shown), and the quadrant analysis showed good agreement of the flow structures and coherent structures when LES was compared with the Direct Numerical Simulations of Coceal et al. (2007) in Nazarian et al. (2018b). Lastly, we compared the TKE profiles obtained with the LES results with the experiment of Brown et al. (2001) and observed good agreement in the

shape of the profiles and TKE above the canyon, while underestimation of TKE within the building levels is seen (Fig. 4). Such underestimation of TKE compared to measurements in the canopy was also reported in Giometto et al. (2016). Additionally, since the exact value of friction velocity was not available in the experimental dataset, the velocity at $3H$ is used for this comparison which may further contributes to the discrepancy. A direct comparison between LES and RANS demonstrates that RANS underestimates TKE even further compared to the wind tunnel results (Sect. 2.2.3).

Second, in order to ensure the accuracy of our LES analysis, the choice of simulation set-ups are rigorously evaluated here, and a series of sensitivity analyses are performed to compare the profiles (time and ensemble-averaged) of mean flow, TKE and velocity covariances based on the 1) geometrical configuration (size and height of the domain), 2) grid resolution, and 3) runtime parameters (spin-up time, sampling frequency, and time averaging interval).

We find the domain height to be critical for both staggered and aligned arrays. The domain size of 4H previously used in the

RANS simulations of Santiago and Martilli (2010) is insufficient for the present LES analysis, as it modifies the vertical profile of Reynolds stresses and accordingly TKE above the building height. Therefore, following a series of sensitivity analyses, $7.4\,H$ is used to ensure that the top boundary condition (i.e. the lack of solution for the entire boundary layer) minimally affects simulation results in the roughness sublayer. Similarly, the choice of domain size (number of buildings in an array) is critical. In the streamwise direction, a sufficient number of buildings should be included in the computational domain to resolve the large

eddies influencing the canopy flow (Inagaki et al., 2012; Coceal et al., 2006). Similarly, Yaghoobian et al. (2014) compared 3x3,





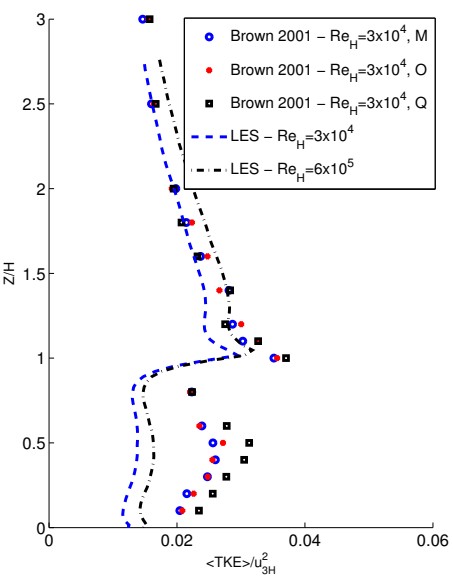

**Figure 4.** Comparison of the TKE profile at the center of the canyon with experimental results of Brown et al. (2001).

5x3, and 5x5 arrays of aligned buildings and found 5x3 to be the best compromise between accuracy and computational cost. In this analysis, we extended the domain to an array of 6x4 aligned cubes, and found insignificant differences in the vertical profile of turbulent parameters. Therefore 5x3 and 6x3 array of cubes are selected for aligned and staggered configurations, respectively. Additionally, the grid resolution of 32 grid cells per H is used following the grid sensitivity analysis done by

Yaghoobian et al. (2014) and Nazarian et al. (2018b) that showed lower grid resolution (such as 0.05H, or 20 grid cells per canopy height H) to be insufficient for resolving the wall flow.

Regarding the runtime calculations, 3 main parameters are evaluated. First the volume-averaged results are monitored throughout the runs, and the spin-up time (i.e. the initial time interval that is discarded in the subsequent analysis) is chosen to be 3 hrs, corresponding to 125 eddy turnover time ($T = H/u_\tau$). This initial time interval is necessary to reach quasi-steady

behavior in TKE, velocity variances, and friction velocity at the surfaces. Second, the choice of sampling frequency is evaluated by comparing the vertical profiles of TKE and Reynolds stresses for 10, 20 and 50 time step sampling frequencies (time step size is 2 seconds). TKE results are influenced when low frequency (50 time steps) is used. However, there is no significant change in the TKE profile between 10 and 20 timesteps, though the computational cost is affected. Hence, results are saved every 20 time steps. The last and most important factor in the runtime parameters is the time-averaging intervals. Coceal et al.

(2006) and Nazarian et al. (2018b) have shown that in order to filter the formation of roll-like circulations over the urban-like configurations, the results should be averaged over a time period of 200-400 eddy turnovers. Therefore, in this analysis, the results are averaged over 6hrs, which corresponds to $250T$.





### 2.2.3 Model Evaluation: Comparison with RANS

Santiago and Martilli (2010) used vertical profiles of flow properties calculated from Reynolds-averaged Navier-Stokes (RANS) simulations of idealized arrays of buildings (setups similar to Sect. 2.2.1) to parameterize the 1-D column model (Sect. 2.1). The RANS model used for the urban canopy parameterization showed good agreement when evaluated against DNS and wind

tunnel results for flow over aligned cube arrays (Santiago et al., 2008; Simón-Moral et al., 2014), and wind tunnel results for canopies of 'vegetation' (Santiago et al., 2013; Krayenhoff et al., 2015). When compared to the Large-Eddy Simulation results (Fig. 5), the streamwise velocity as well as Reynolds stress at the building height calculated in RANS shows agreement with the LES results. For the vertical profile of Reynolds stress ($\langle \overline{u'w'} \rangle$), the aligned configuration results in a better agreement within the canyon, while the above-canopy results are mainly dominated by the domain height (which is set at 4H in RANS,

significantly lower than 7.4H in LES). However, when the vertical variation of normalized turbulent kinetic energy ($\langle \overline{k} \rangle / u_\tau^2$) is compared to wind tunnel experiments (not shown) and LES (Fig. 5), RANS substantially underestimates turbulent kinetic energy in the urban canopy layer. Similar behavior is previously reported when the distribution of TKE obtained by LES and RANS models are compared with the measurements in a realistic urban configurations (Antoniou et al., 2017) and wind tunnel experiments (Xie and Castro, 2006). Additionally, Krayenhoff et al. (2019) suggested that the 1-D model of Santiago and Mar-

tilli (2010) contributed to underestimation of the venting in UCL, and the discrepancy has been traced to turbulent length scales derived from the RANS simulations. These new findings, and the recent advancements in the high-performance computing, motivate a revisitation of these parameterizations with a more accurate flow model such as LES that has been shown to be superior in representing the turbulent flow statistics (Xie and Castro, 2009; Salim et al., 2011; Gousseau et al., 2011).

## 3 Results and Discussions

### 3.1 Large Eddy Simulations: Vertical Profile of Mean Flow and Dispersive Stress

Figure 6 displays the vertical profiles of flow parameters ($\langle \overline{u} \rangle / u_\tau$, $\langle \overline{k} \rangle / u_\tau^2$, and $\langle \overline{u'w'} \rangle / u_\tau^2$) spatially averaged within the roughness sublayer for two urban configurations (aligned and staggered) and varying urban density ($\lambda_p$). It is evident that the flow profiles are significantly influenced by the urban configuration. Overall, average wind speed, and consequently the turbulent momentum flux and TKE, are significantly larger for aligned array of cubes with streamwise flow aligned with urban

street canyon. However, it is worth noting that in real cities, the aligned configuration with $0°$ wind angle may not be most representative of flow field. Real cities experience a range of wind directions relative to the street grid, and many cities do not have a grid but rather streets of many orientations. Our simulations (similar to many 'urban' CFD simulations) represent buildings with two street directions oriented perpendicular to each other, with streamwise flow oriented perpendicular to one set of building faces. The aligned version of this setup represents a special case relative to real cities: those scenarios where

wind direction is aligned with one of the two street directions. The staggered version of this setup, conversely, presents no major corridors (i.e., streets) aligned with the wind that do not include building drag. As such, we believe that the staggered configuration better represents the impacts of real cities on urban canopy flow under a variety of wind directions. Any choice



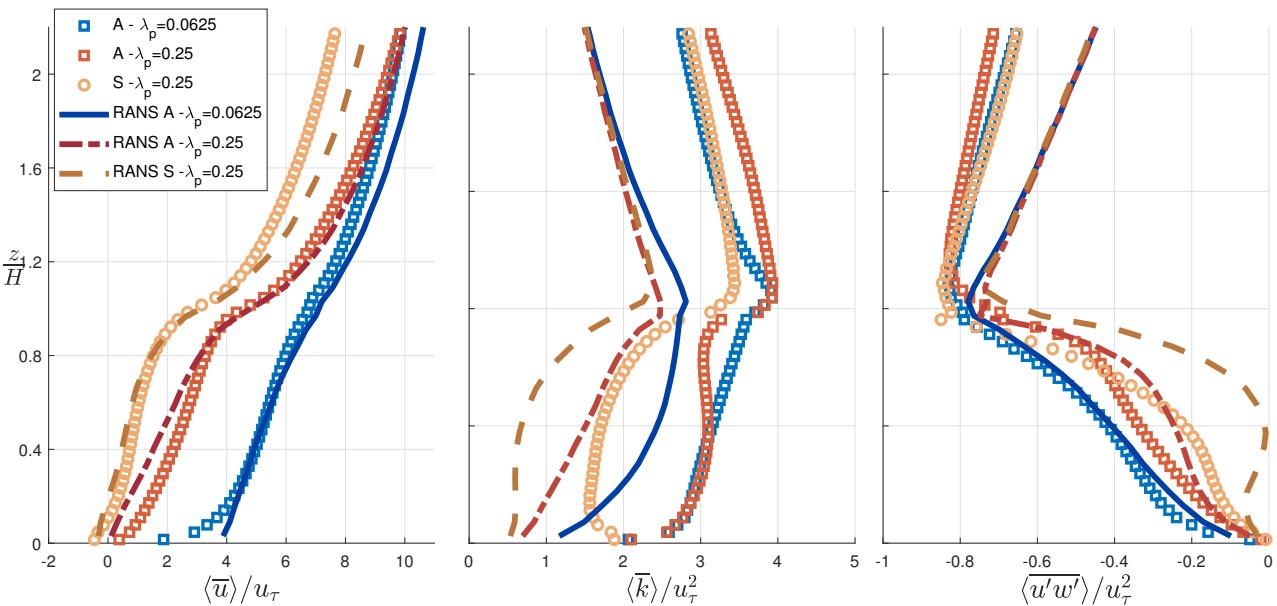

**Figure 5.** Vertical profile of velocity ($\langle\overline{u}\rangle/u_\tau$), TKE ($\langle k\rangle/u_\tau^2$), and turbulent momentum flux ($\langle\overline{u'w'}\rangle/u_\tau^2$) compared with the RANS simulation (Santiago and Martilli, 2010) at different urban densities. "A" and "S" correspond to aligned and staggered arrays of buildings.

here is a simplification of reality, and the choice of a regular staggered array provides a closer approximation to average conditions in real cities in our estimation.

Another investigation made here is regarding the significance of dispersive fluxes in urban canopy parameterizations. In the formulation of the multi-layer urban canopy model, the dispersive transport processes are neglected so far (Santiago and
5    Martilli, 2010; Krayenhoff et al., 2015), while in fact they are non-negligible in many real urban configurations (Giometto et al., 2016). The variability of the spatially-averaged dispersive stress obtained from LES for varied urban configuration and packing density and the contribution of $\langle\tilde{u}\tilde{w}\rangle$ to the total turbulent momentum flux ($\langle\overline{u'w'}\rangle + \langle\overline{u'w'}\rangle$) is represented in Fig 7. It is observed that the dispersive stress can in fact be in the same order of the total momentum flux. Hence, given the importance of the dispersive term in the momentum budget, the subsequent analysis seeks to represent the effects of dispersive motions in
10   the column model by driving the parameterization from the 3D results of $\langle\tilde{u}\tilde{w}\rangle$ together with $\langle\overline{u'w'}\rangle$ (Sect. 3.3).



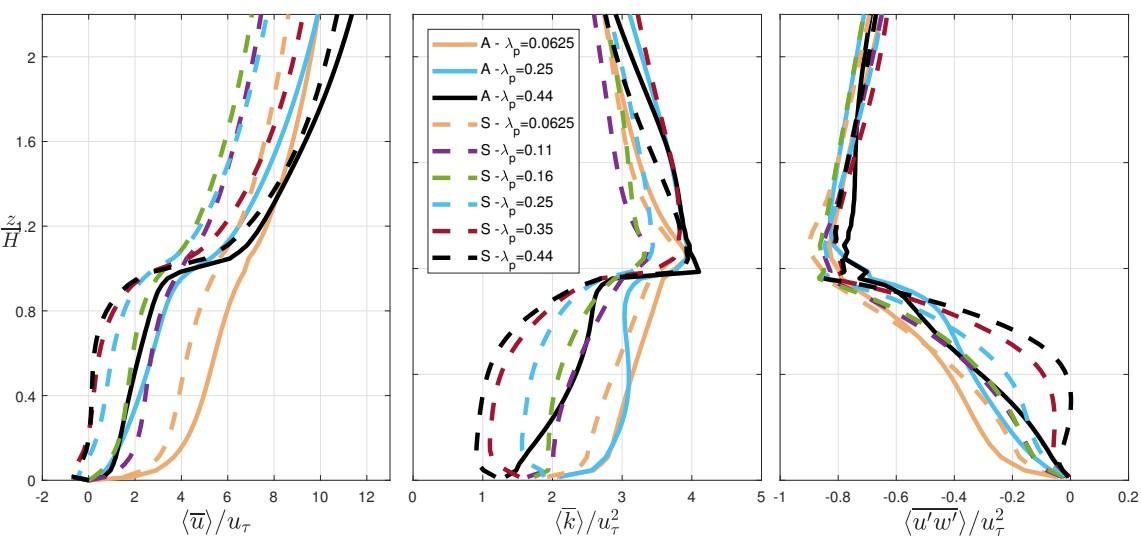

**Figure 6.** Vertical profiles of normalized velocity ($\langle \overline{u} \rangle / u_\tau$), turbulent kinetic energy ($\langle \overline{k} \rangle / u_\tau^2$), and turbulent momentum flux ($\langle \overline{u'w'} \rangle / u_\tau^2$) obtained from LES results, spatially- and time- averaged for different $\lambda_p$ and configurations. $\langle \overline{k} \rangle$ from the LES results are calculated based on the turbulent variances at the resolved scale as well as modeled subgrid-scale TKE. "A" in this graph indicates "Aligned" configuration, while "S" stands for "Staggered".

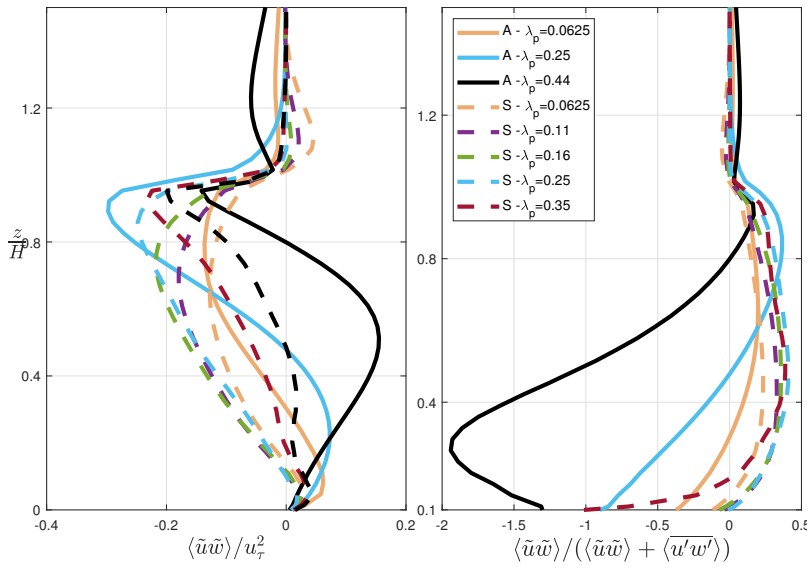

**Figure 7.** Vertical (spatially and temporally averaged) profiles of a) normalized dispersive stress $\langle \tilde{u}\tilde{w} \rangle / u_\tau^2$ and b) the contribution of dispersive stress to the total turbulent momentum flux $\langle \tilde{u}\tilde{w} \rangle / (\langle \overline{u'w'} \rangle + \langle \overline{u'w'} \rangle)$. "A" in this graph indicates "Aligned" configuration, while "S" stands for "Staggered".



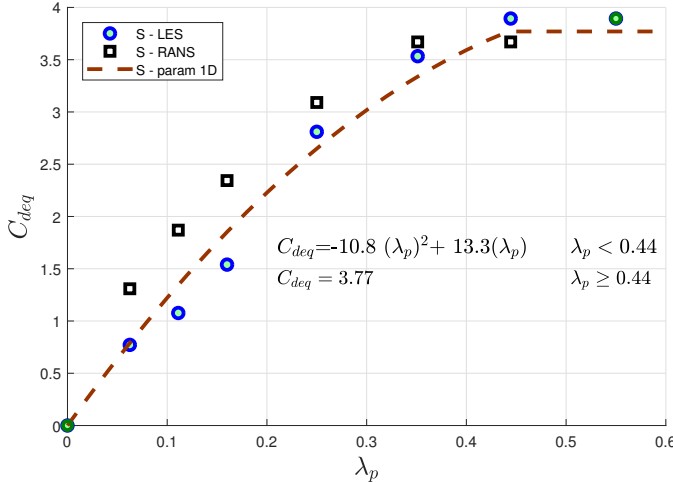

**Figure 8.** Variation of $C_{deq}$ with urban packing density $\lambda_p$ for the staggered configuration, and compared with the RANS results. The fitted line indicates the parameterization proposed for the 1-D model. Additional data points (in green) are added for parameterization corresponding to $\lambda_p = 0$ (zero building-induced drag) and $\lambda_p > 0.44$ (where the drag coefficient reaches a constant value for high urban packing densities).

## 3.2 Drag Parameterization

It is known that the sectional drag coefficient depends on the packing density and the configuration of the array with a strong dependency with height, such that $C_d = C_d(z, \lambda_p)$ (Macdonald, 2000; Santiago et al., 2008; Santiago and Martilli, 2010). However, as indicated by Santiago and Martilli (2010), height-dependent parameterization of drag coefficients is challenging
5   due to the high variability of $C_d$ close to the ground due to small $\langle \overline{u} \rangle$ as well as the lack of experimental information on the vertical profiles of this property inside the urban canopy. Accordingly, Santiago and Martilli (2010) proposed the following calculation of sectional drag coefficient that is constant with height in the urban canyon, considering that when integrated in the whole urban canopy, the drag force must be equal to that computed by the CFD simulations.

$$C_{deq} = \frac{\frac{-1}{\rho h} \int_0^h \Delta \langle \overline{p(z)} \rangle dz}{\frac{1}{h} \int_0^h \langle \overline{u(z)} \rangle | \langle \overline{u(z)} \rangle | dz} \qquad (10)$$

10   Following this method, the drag coefficient parameterization using the LES results is shown in Fig. 8. $C_{deq}$ is computed by means of the ratio between the horizontally averaged mean pressure deficit around an obstacle and the square of the horizontally averaged mean velocity around the obstacle (Eq. 10). $C_{deq}$ depends on the configuration (aligned or staggered) and packing density ($\lambda_p$) of the array shown here. The $Cd_{eq}$ values from LES simulations are in good agreement with the RANS results, but as previously demonstrated by Simón-Moral et al. (2014), RANS tend to overestimate the value of $C_{deq}$.





### 3.3 Length Scale Parameterization

In this section, the length scales obtained from the spatially-averaged LES results and the $k-l$ turbulence closure theory for the urban canopy parameterization is discussed. Following the discussion in Sect. 3.1, the turbulent length scale $C_k l_k$ is calculated using total momentum fluxes that include turbulent dispersive flux $\langle \tilde{u}\tilde{w} \rangle$ (Eq. 11b), as opposed to only considering

the Reynolds stress $\langle \overline{u'w'} \rangle$ (Eq. 11a). Note that in the column model, $C_k l_k$ ($l_\epsilon/C_\epsilon$) is parameterized instead of $l_k$ ($l_\epsilon$) in order to avoid the uncertainties regarding the values of $C_k$ ($C_\epsilon$) proposed in the literature.

$$\langle \overline{u'w'} \rangle = -C_k l_{kt} \langle \overline{k} \rangle^{1/2} \frac{\partial \langle \overline{u} \rangle}{\partial z} \tag{11a}$$

$$\langle \overline{u'w'} \rangle + \langle \tilde{u}\tilde{w} \rangle = -C_k l_{kM} \langle \overline{k} \rangle^{1/2} \frac{\partial \langle \overline{u} \rangle}{\partial z} \tag{11b}$$

Figure 9 shows the vertical profile of turbulent length scale calculated for varied urban densities (left) and the canopy-averaged

length scale obtained from RANS and LES with/without considering the dispersive term (right). Two observations are made. First, we observe that the turbulence length scale is larger for the LES results within the building canopy, specifically when dispersive stress is included. This is further explained by the significant difference in the TKE profile between LES and RANS shown in Fig. 5. Second, the length scale calculated using LES does not vary monotonically with urban packing density ($\lambda_p$) but rather follows the behavior of roughness length (Grimmond and Oke, 1999). This can be explained due to the varying

flow regimes from the isolated ($\lambda_p = 0.0625$) to wake interference ($\lambda_p = 0.25$) and skimming ($\lambda_p = 0.44$) flow. As noted by Grimmond and Oke (1999), "as the density increases so does the roughness of the system, but a point comes where adding new elements merely serves to reduce the effective drag of those already present due to mutual sheltering ... This reduces the effective height of the canopy for momentum exchange". Accordingly, we observe that the drag coefficient (Fig. 8) plateaus with increasing density. Similarly, the non-monotonic behavior in LES-derived turbulent length scales (that resolve the turbulent

flux of momentum and energy across larger scales of motions as opposed to RANS) can be attributed to different flow regimes. The LES results suggest that the largest scales of turbulence (i.e., the most turbulent organization) are produced in the wake interference regime. As the turbulent length scale is a measure of the efficiency of vertical transport, the higher $l_k$ values indicate higher vertical transport of momentum (including turbulent and dispersive) for the same TKE and vertical flow gradient. For intermediate $\lambda_p$, mainly in the wake interference regime, the presence of the buildings favor the formation of organized

motions, likely at the scale of the buildings, that enhance the vertical transport. For higher $\lambda_p$ values, in the skimming flow, this movements are suppressed, and for isolated buildings are less strong. Given the time-averaged representation of the turbulent field, RANS is not able to reproduce these effects, resulting in the monotonic decrease in derived turbulent length scale with urban density.

To assess the dissipation length scale $l_\epsilon/C_\epsilon$, Eq. 6 is used assuming that the dissipation is only happening at the subgrid scale,

and therefore is correlated with the TKE-SGS (Maronga et al., 2015). Figure 10 (left) demonstrates the vertical profile of $l_\epsilon/C_\epsilon$ for staggered urban configurations with variable packing density. The vertical profiles of $l_\epsilon/C_\epsilon$ exhibit similar characteristics compared to the RANS results of Santiago and Martilli (2010): inside the canopy, length scale is mostly constant with height (specifically for $\lambda_p \geq 0.25$) as it is controlled by the shear layer ($h-d$), above the canopy the dissipation length scale increases



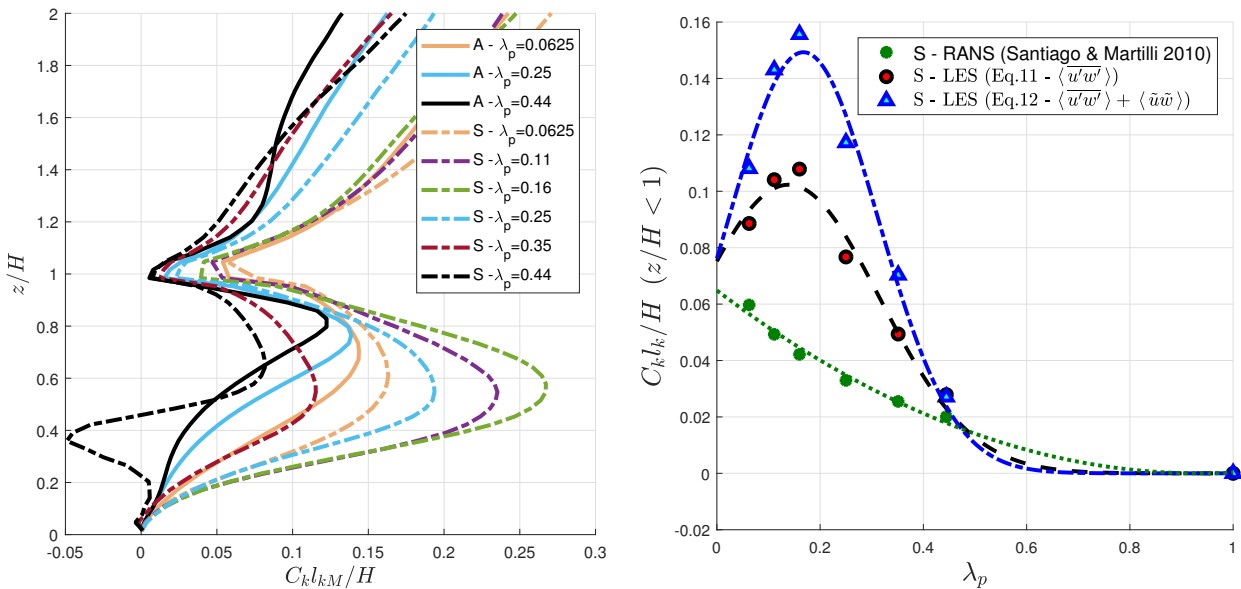

**Figure 9.** Left: Vertical profiles of turbulent length scale calculated using LES simulations with the dispersive stress included for aligned (A) and staggered (S) urban configurations. Right: Variation of normalized turbulent length scale averaged within the building canyon ($z/H < 1$) with plan area density ($\lambda_p$) for 1) RANS simulation (Santiago and Martilli, 2010), 2) LES simulation without dispersive stress included, and 3) LES simulation with dispersive stress included.

with height, and the lower values of $l_\epsilon/C_\epsilon$ close to the ground (particularly for lower $\lambda_p$) and at building height corresponds to the locations of maximum dissipation in the urban canopy. This is likely due to the fact that dissipation depends only on small scale motions, and therefore is less affected by larger structure induced by the presence of the buildings.

Three different zones is then defined consistent with Santiago and Martilli (2010) and Krayenhoff et al. (2015) to parame-

5 terize $L_\epsilon/C_\epsilon$: a) inside the canopy ($z/H < 1$), $L_\epsilon/C_\epsilon$ is assumed constant with height, b) well above the canopy ($z/H > 1.5$), where the behaviour of $L_\epsilon/C_\epsilon$ is linear and the slope varies with $\lambda_p$, and c) in the zone of transition ($1 \leq z/H \leq 1.5$) between the two previous zones:

$$L_\epsilon/C_\epsilon = \alpha_1(H - d) \qquad z/H < 1 \tag{12a}$$

$$L_\epsilon/C_\epsilon = \alpha_1(z - d) \qquad 1 \leq z/H \leq 1.5 \tag{12b}$$

10 $$L_\epsilon/C_\epsilon = \alpha_2(z - d_2) \qquad z/H > 1.5 \tag{12c}$$

where $\alpha_1 = 4$ is the revised value computed for all $\lambda_p$ cases in LES, the displacement height ($d$) parameterization is taken from Krayenhoff et al. (2015) and Simón-Moral et al. (2014) as

$$d(\lambda_p) = H\lambda_p^{0.15} \tag{13}$$



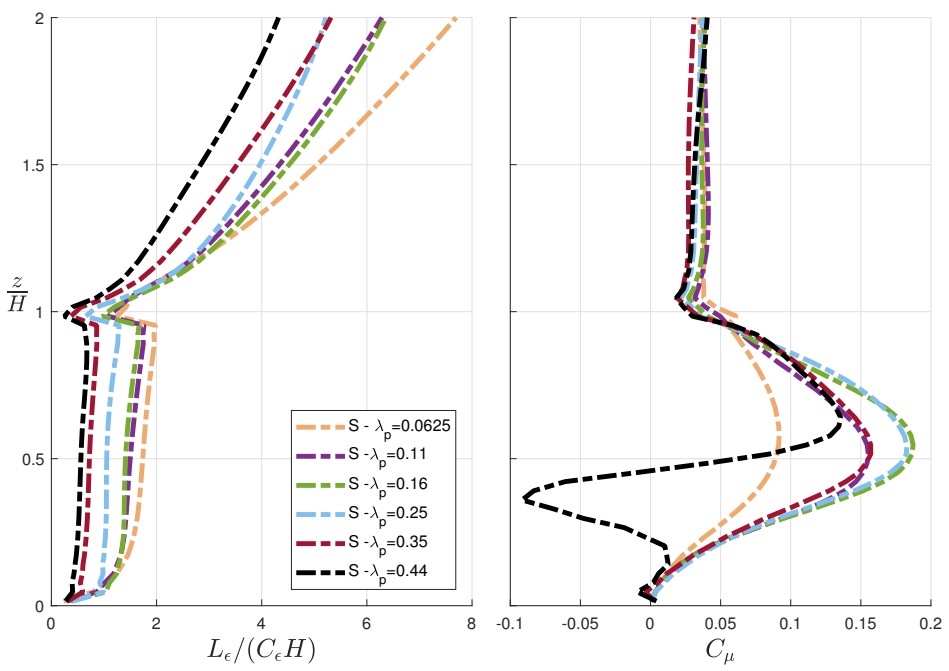

**Figure 10.** Left: vertical profiles of dissipation length scale ($L_\epsilon/C_\epsilon$) for staggered (S) urban configuration and varying urban packing density ($\lambda_p$). $L_\epsilon/C_\epsilon$ is calculated based on dissipation at the subgrid scale (Maronga et al., 2015) and Eq. 6. Right: vertical profiles of model constant for turbulent viscosity ($C_\mu$) calculated based on Eq. 7.

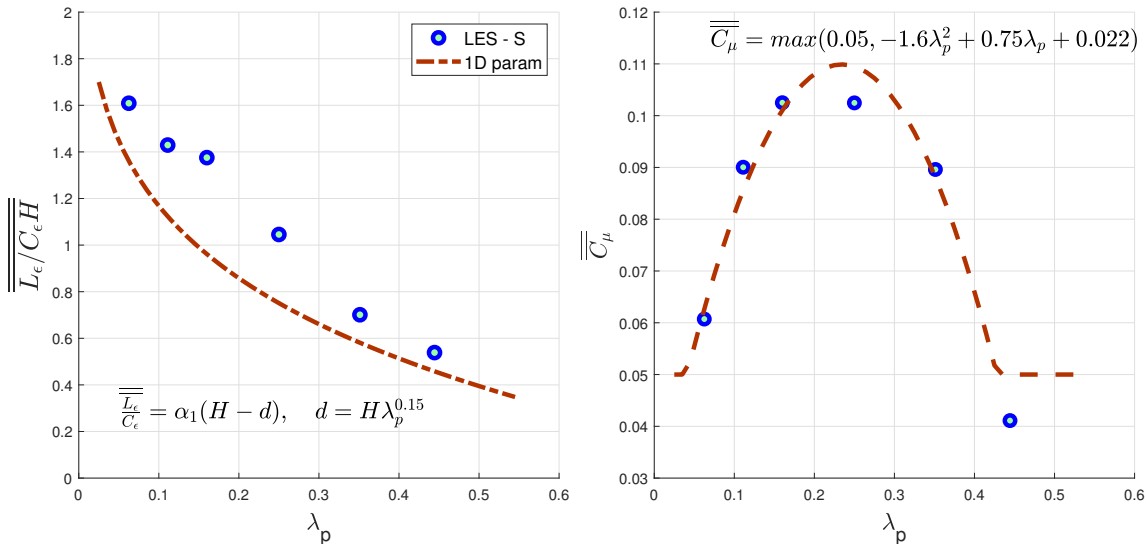

**Figure 11.** Variation of normalized dissipation length scale ($L_\epsilon/C_\epsilon H$) and model constant for turbulent viscosity ($C_\mu$) averaged within the canopy ($z/H <= 1$) with plan area density ($\lambda_p$). The results are obtained using the staggered urban configuration (Fig. 3).





and finally $\alpha_2$ and $d_2$ are parameterized as

$$\alpha_2(\lambda_p) = \min(5, \max(2, 1.3\lambda_p^{-0.45})) \,, \tag{14a}$$

$$d_2(\lambda_p) = 1.5H(1 - \frac{\alpha_1}{\alpha_2}) + d(\lambda_p)\frac{\alpha_1}{\alpha_2} \,. \tag{14b}$$

Note that $\alpha_1$ (in-canopy) does not vary significantly with urban density while $\alpha_2$ (slope of $L_\epsilon/C_\epsilon$ above $Z/H = 1.5$) is a function of $\lambda_p$. Additionally, the parameterization of the dispersive length scale below building height ($\alpha_1$) is slightly underestimated compared to the LES results to account for the localized maximum of dissipation close to the ground specifically for low urban densities (Fig. 11).

Comparing the turbulent ($C_k L_k$) and dissipation ($L_\epsilon/C_\epsilon$) length scales (Eq. 7), however, we find that the assumption of constant $C_\mu$ in the canopy does not hold in the LES results. Figure 10-right demonstrates the vertical profile of $C_\mu$ calculated based on Eq. 7 and we observe the variability in in-canopy $C_\mu$ with $\lambda_p$. Accordingly, in addition to the dissipation length scale parameterization provided for the 1-D model, the $C_\mu$ value in the canopy is also parameterized based on $\lambda_p$ (Fig. 11-right), while above the canopy, $C_\mu = 0.05$ for all urban densities:

$$C_\mu = \begin{cases} max(0.05, -1.6\lambda_p^2 + 0.75\lambda_p + 0.022) & z/H \leq 1 \\ 0.05 & z/H > 1 \end{cases} \tag{15}$$

### 3.4 Assessment of One-Dimensional Column model with LES and RANS Results

The drag coefficient (Sect. 3.2) and length scales (3.3) parameterizations derived from LES results are used to update the multi-layer (1-D) urban canopy model of Santiago and Martilli (2010) and evaluated here against 1) RANS-derived multi-layer (1-D) model, 2) 3D LES results with idealized configuration (present study) and 3) LES results with realistic urban configurations (Giometto et al., 2017).

Figure 12 demonstrates the vertical profiles of horizontally-averaged velocity, turbulent kinetic energy, and turbulent momentum flux calculated with the 1-D (multi-layer) model and compared with the LES results. The 1-D model results are calculated using previous parameterizations with RANS (Santiago and Martilli, 2010) as well as the updated LES parameterizations for $\lambda_p = 0.0625, 0.25$, and $0.44$. The vertical profile of $\langle \overline{u'w'} \rangle / u_\tau^2$ obtained with the updated (LES) 1-D model shows improvement for all studied $\lambda_p$ cases. For horizontally averaged $\langle \overline{u} \rangle / u_\tau$ and $\langle \overline{k} \rangle / u_\tau^2$, however, the performance of the model is dependent on the $\lambda$ value. Overall, the prediction of the horizontally-averaged velocity is improved compared to 1D-RANS, particularly at the canopy level. However, significant underestimation of wind speed is seen at the higher urban density. TKE profiles, on the other hand, are overestimated for $\lambda_p = 0.0625$ while significantly improved for other cases. Additionally, despite the improvements with the new parameterization, the TKE close to the ground is still substantially underestimated for high $\lambda_p$ cases, indicating there is underestimation of vertical turbulent transport deep in the canopy. This could be traced back to the parameterization of the TKE transport in the multi-layer model that assumes the same diffusion coefficient ($K_m$) for momentum and turbulent equation, which does not hold in the LES results (not shown).



**Figure 12.** Comparison of the vertical profiles of velocity ($\langle \overline{u} \rangle / u_\tau$), turbulent kinetic energy ($\langle \overline{k} \rangle / u_\tau^2$), and turbulent momentum flux ($\langle \overline{u'w'} \rangle / u_\tau^2$) obtained with the multi-layer (1-D) model with RANS and LES parameterization with the LES results for various $\lambda_p$.



Figure 13 further demonstrates the root-mean-square error (RMSE) of horizontally-averaged velocity ($\langle \overline{u} \rangle / u_\tau$), turbulent kinetic energy ($\langle \overline{k} \rangle / u_\tau^2$), and turbulent momentum flux ($\langle \overline{u'w'} \rangle / u_\tau^2$) compared with the LES results. RMSE is calculated for $z = 0 - 3H$ for all $\lambda_p$ cases studied here. It can be seen that the new parameterizations with LES (depicted in dark blue) represents an overall improvement compared to the previous multi-layer model derived from the RANS results. The RMSEs for $\langle \overline{u'w'} \rangle / u_\tau^2$ and $\langle \overline{k} \rangle / u_\tau^2$ are substantially lower in the updated multi-layer model with LES-derived parameters and formulations. For high-$\lambda_p$ cases, however, the new parameterization underperforms in predicting the wind flow.

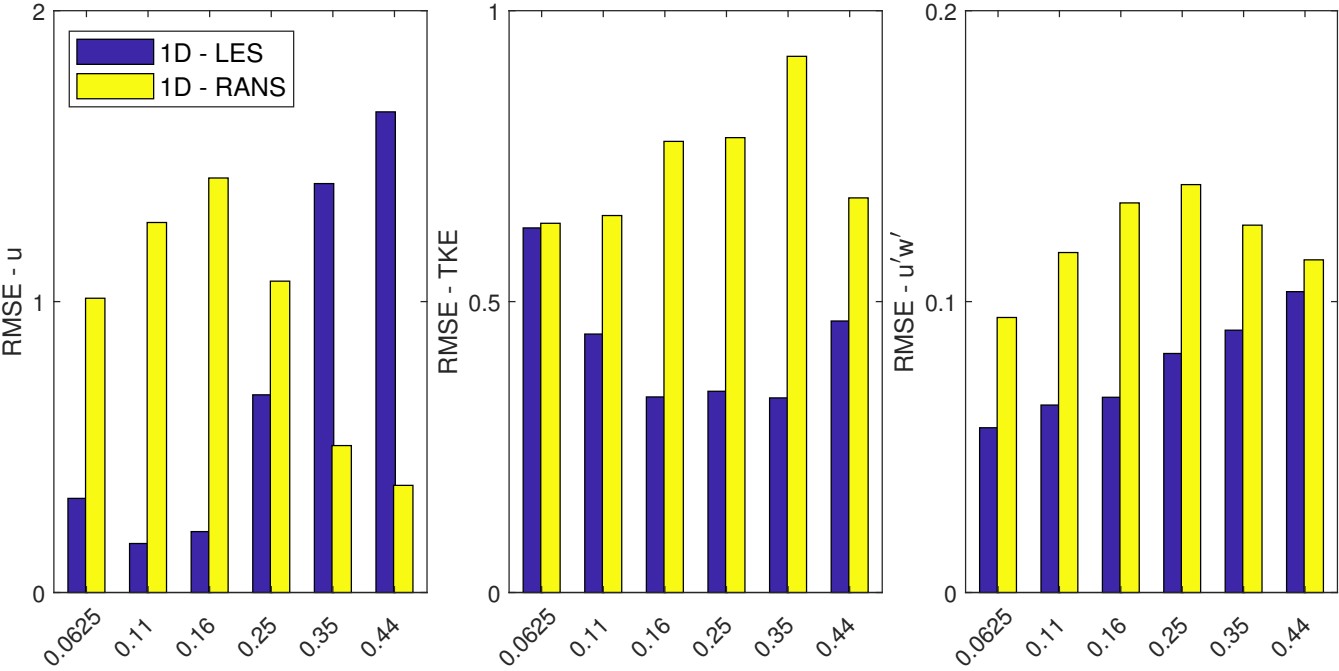

**Figure 13.** Root-mean-square-error calculated for vertical profiles of velocity ($\frac{u}{u_\tau}$), TKE ($\overline{k}/u_\tau^2$), and Reynolds stress ($u'w'$)$/u_\tau^2$. The RMSE values are calculated using 1-D models with LES and RANS parameterization up to 3H.

Lastly, the 1-D multi-layer model is compared with LES results of Giometto et al. (2017) that is conducted for a realistic urban neighborhood (Vancouver-Sunset) in BC, Canada. The neighborhood characteristic in the modeled urban canopy subset (indicated as S1 in Giometto et al. (2017)) is $\lambda_p = 0.34$ and average building height is $6.6\,\mathrm{m}$. The studied case in Fig. 14 represents a configuration without trees (given the fact that tree parameterization was not the focus of the current study). We observe that the updated parameterizations in the 1-D multi-layer model result a substantial improvement compared to Santiago and Martilli (2010), specifically for the vertical profile of turbulent kinetic energy as well as wind speed above the building height. However, underestimation of wind speed and Reynolds stress in the street canopy is observed which is likely attributed to the building configuration and wind direction considered in the realistic LES simulations. In Giometto et al. (2017), the urban configuration resembles evenly-spaced aligned buildings with wind direction aligned with one of the primary street directions. This results in a relatively linear profiles of wind speed and Reynolds stress in the canopy, which as discussed



before, only represent one realization of urban canopy flow. Nonetheless, this demonstrates the need for assessing urban canopy parameterization with various urban configuration and wind direction in the future. Additionally, underestimation of TKE deep in the canopy is seen again, further indicating that the current parameterization of the turbulent transport in the urban canopy is not adequate to determine $\langle \overline{k} \rangle$ at the ground level, particularly in higher $\lambda_p$ cases.

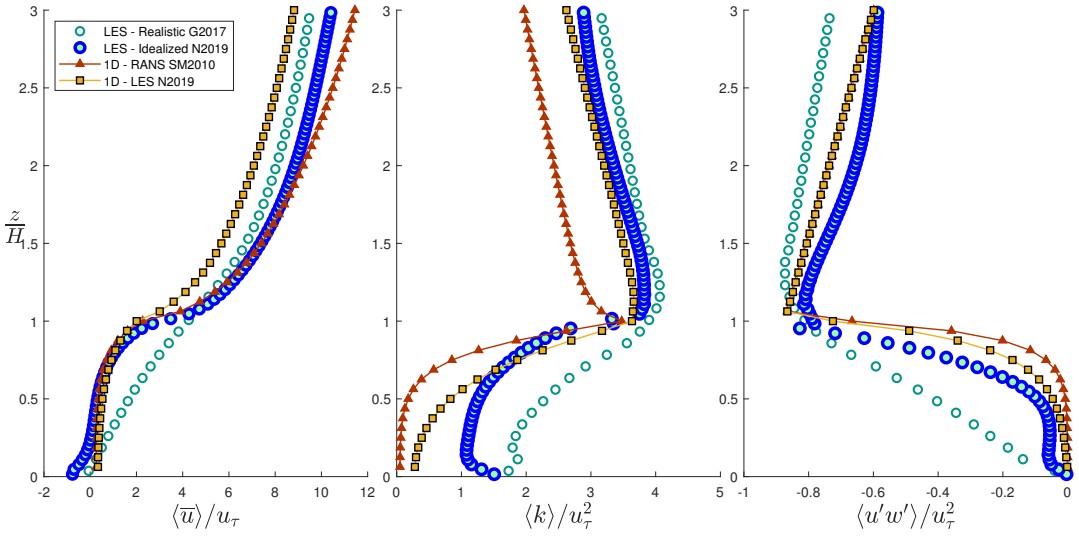

**Figure 14.** Comparison of vertical profile of velocity ($\langle \overline{u} \rangle / u_\tau$), turbulent kinetic energy ($\langle \overline{k} \rangle / u_\tau^2$), and turbulent momentum flux ($\langle \overline{u'w'} \rangle / u_\tau^2$) obtained with multi-layer (1-D) model with the LES simulations of Giometto et al. (2017) for realistic urban configurations.

## 4  Summary and Conclusions

The present study focused on updating the urban canopy parameterizations of drag coefficient and turbulent length scales using Large Eddy Simulations (LES) results, which is shown to be a superior numerical model for resolving the turbulent flow field compared to Reynolds-Averaged Navier Stokes (RANS) previously used in multi-layer UCMs (Santiago and Martilli, 2010).

The detailed analyses of spatially-averaged turbulent field in urban configurations revealed the following: 1) LES results exhibit a significantly higher transport of TKE into the lower canopy compared to RANS; 2) dispersive fluxes are not negligible in the urban canopy, particularly in higher urban packing densities; and 3) the ratio between turbulent and dispersive length scale (commonly described by the model constant $C_\mu$ in multi-layer models) is not constant with $\lambda_p$ at the canopy level. These findings motivated the revision of the UCPs to include dispersive fluxes and further parameterize turbulent length scale ($C_k l_k$) in addition to dispersive length scale ($l_\varepsilon / l_\varepsilon$) through the parameterization of model constant $C_\mu$.

We demonstrated that using LES results as the basis for parameterization as well as the inclusion of dispersive stress improves the performance of the multi-layer model, such that spatially-averaged profiles of flow, and consequently the turbulent exchange in the urban canopy in realistic neighborhoods, can be predicted more accurately. However, spatially-averaged turbulent kinetic





energy, $\langle \overline{k} \rangle$, is still underestimated close to the ground for high $\lambda_p$ values due to the underestimation of turbulent transport deep in the canopy. Preliminary analyses of turbulent transport in this study (not shown) reveal that the K-theory assumption that diffusion coefficient $K_m$ is the same for TKE and momentum equations (i.e. $K_m = -\frac{\langle u'w' \rangle}{\partial \langle \overline{u} \rangle / \partial z} = -\frac{\langle \overline{k}'w' \rangle}{\partial \langle \overline{k} \rangle / \partial z}$) does not hold in the LES results. Accordingly, future work should revisit the multi-layer model formulations to assess 1) the parameterization of

turbulent transport term in the 1-D TKE equation (Eq. 5) and 2) distinction between the diffusion length scale of momentum and TKE.

    Further analysis is also needed to fully evaluate the effects of idealized configurations in parameterization and assess the impact of variable building heights and wind directions on turbulent length scales and drag parameterization. Lastly, the current study focused on the momentum exchange without considering the role of thermal forcing on turbulent length scales. Updated

parameterization of thermal effects (previously investigated by Krayenhoff et al. 2019) can also be evaluated using LES results.

**Author Contribution**

NN, ESK, and AM collectively developed and planned the study. AM developed the initial one-dimensional vertical diffusion model and ESK and AM have continued the parameterization of 1-D model to include radiation and trees. NN ran the Large-Eddy Simulations and modified the model based on the updated parametrizations of length scales. NN carried out the result

analyses and wrote the paper with significant input and critical feedback from ESK and AM.

**Acknowledgements**

Author Krayenhoff was supported by NSF Sustainability Research Network (SRN) Cooperative Agreement 1444758 and NSF SES-1520803, as well as an NSERC Discovery grant. Author Nazarian acknowledges the funding received from National Research Foundation Singapore under its Campus for Research Excellence and Technological Enterprise programme in the

initial phases of this analysis. The authors thank Dr. Marco Giometto (Columbia University) for sharing the LES data of realistic configurations and acknowledge discussions with Prof. Andreas Christen (Uni Freiburg) and Dr. Andres Simon-Moral (NUS).

**Code and data availability**

The source code and the supporting data of the updated 1D Multi-layer Urban Canopy Model (MLUCM) v1.0.0 is publicly

available at https://github.com/nenazarian/MLUCM under GPL 3.0 licence: https://opensource.org/licenses/GPL-3.0 (last access: May 2019) and can be downloaded from https://www.zenodo.org/ with DOI: 10.5281/zenodo.3464711.

**Competing Interest**

The authors declare that they do not have any conflict of interest.



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
