# Peer review of "A One-Dimensional Model of Turbulent Flow Through 'Urban' Canopies: Updates Based on Large-Eddy Simulation"

_Geoscientific Model Development, 2019_

## Referee Comment (RC1) · Anonymous Referee #1 · 15 Nov 2019

General considerations

In this contribution, the authors use high-resolution LES simulations over an idealized urban-type surface configuration to assess the turbulence characteristics and exchange efficiency for neutral stratification. Results are used to improve a number of scales and parameters, which form the basis for a multi-layer urban surface exchange parameterization for meso-scale atmospheric models and had previously been determined based on RANS-type simulations (with similar surface configuration). A number of configurational improvements (e.g. minimum domain size) are discussed, and also the inclusion of dispersive stresses is emphasized.

The paper is sound in its analysis (sometimes a little too optimistic in its conclusions, though) and the experimental plan, but not very carefully written (see the long list of minor comments). Also, the language could be improved (I have only indicated the obvious language errors). I have got two 'rather major' comments, of which the first is easily addressed, and the second may help to extend the speculations on why even the LES-based parameterization fails to reproduce some of the characteristics in real-building urban configurations. Once the comments are properly addressed, the paper can certainly be recommended for publication in GMD.

Major comments

1) The paper is based on an earlier finding of the authors (or some of them) that the BEP-tree underestimates vertical exchange of momentum and energy in the canopy (p3, l.25). This is essentially concluded from comparison to one experiment in Vancouver. I trust that the authors have also assembled other evidence from earlier validations of the model. It would be more convincing to support the motivation for the paper to briefly summarize this additional evidence – or at least make a reference to where it can be found.

2) P5, l.14 (u and w...): by neglecting the lateral component (and the corresponding covariances) the authors only consider frictional stress and not directional stress (v'w' – including its dispersive contribution). At the top of the canopy – and depending of course on the wind direction relative to the orientation of the canyon axis – directional stress might be quite important as well. Can the authors comment on this?

Minor comments

P2, l.7 ...from the street-scale to synoptic scales.

P2, l.13 'is being treated' may be better

P3, l32 I'd use 'hypothesize' rather than speculate

P6, l. 13 is calculated..

Eq 8 I think, the ending | of the 'absolute value' () is missing here. Also, the rhs has dimensions of ms-2 (which corresponds to the usual acceleration due to the pressure gradient force), but the lhs is from eq. (1), which is multiplied with the density – so the rhs needs to be multiplied by density as well.

P7, l. 5 the simulation domain

P7, l. 17 the canyon height

P7, l.18 uniform grid resolution: but what it is? Also 0.03H in the horizontal?

P8, l.5 Fig. 4: is this the aligned or the staggered configuration from which the modeled data is averaged?

P8, l. 9 compared to the wind tunnel results: it has not yet been mentioned that the Brown et al. study has used wind tunnel experiments.

P8, l. 17 7.4H is used to ensure...: so, is Fig. 4 presenting results with the 7.4H domain height? Same (below) for the number of obstacles in the domain. Both should be mentioned in the caption of Fig. 4.

P8, l.17 what is 'the lack of solution [for the entire BL]'?

Fig. 4 legend: what are the letters 'M', 'O' and 'Q' referring to? Also, in the text, k is used for TKE – so, it should be so in the figure.

Fig. 5 caption: 'compared with'. WHAT is compared to RANS? (I assume the PALM output...but needs to be mentioned)

P10, l. 17 a more accurate flow model: but still, Fig. 4 suggests that PALM underestimates TKE (by a factor of 2 or so in the mid- canyon). Can the authors comment on this?

Figs. 5 and 6 the two figures share the same information for the three cases of Fig. 5 but they have different colors for the same configuration (what is yellow/brownish dots

in Fig. 5 is a green dashed line in Fig. 6. It would be very helpful if these colors were the same.

P11, l 7 the total flux is given as <overbar(u'w')> + <overbar(u'w')>. One of the two terms should be the dispersive flux. (same in the caption of Fig. 7)

Fig. 7, caption 'a)' and 'b)': the panels do not have the labels a) and b). Also, it is a little disturbing, that the two panels do not have (exactly) the same vertical axis.

Eq (10) one is inclined to assume that 'h' in the integrals corresponds to the canopy height - but this has been denoted 'H' before. This must be consistent. Further, the notation: $C\_deq$ is what? The sectorial drag coefficient or an 'equilibrium' drag coefficient? Also, the notation 'DELTA  ||' for the horizontally averaged mean pressure deficit seems to be wrong. It should rather be something like <DELTAp(z)> (with an overbar on p)

P14, l.13 are in good agreement: first, the authors probably want to refer to Fig. 8 (which is never mentioned in the text). Second, this figure shows that the 'good' agreement is rather qualitative than quantitative for small lambdas (e.g. for lambda=0.11, the difference in $C\_deq$ is almost a factor of two...)

P14, l.3 are discussed

Eq 11 in this equation, and together with the text, the length scale $l\_kM$ is introduced without explicitly mentioning it. In eq (7), this was $l\_k$ (as it is in the text), and in (11a) it is $l\_t$. All this must be carefully introduced, so that the reader knows what is 'generic' (I assume $l\_k$) and what is particular.

Fig.9 legend refers to eqs. 11 and 12, but should be 11a and 11b.

P14, l. 32 the length scale...

P14, l.33 h-d: I assume 'd' is the zeroplane displacement height, but this has not been introduced so far. And again, canopy height is now 'h' rather than 'H' as previously.

P15, l.1 correspond to

P15, l. 4 are then defined

P15, l.5 L_eps is now capitalized (also in Fig. 9 and its caption), while it was 'l_eps' in eqs (6) and (7) and on p15, l.1 (and last paragraph of p14). All this must be consistent throughout the paper.

Fig. 11 what are the 'double overbars' referring to here? (eq. (15), for example, does not have it...)

P17, l.21 Fig. 12 shows (does not demonstrate)

P17, l.27 especially at the canopy level: if you mean with this, z/H=1 exactly, this is probably not true (at z/H=1, for all three densities, the red line is closer to the blue circles. So, where exactly do you mean?)

P17, l.32 what is the 'turbulent equation'?

Fig.14, caption in the legend, new abbreviations are introduced ('G2017', 'N2019' etc. which need to be explained in the caption).

P19, l.11 result in a substantial

P19, l.16 results in relatively

P20, l.2 parameterizations

P20, l9 of the spatially...

P20, l.14 dissipation length scale

P21, l.2 assumption that the diffusion...

---

## Short Comment (SC1) · 15 Nov 2019

Dear authors,

in my role as Executive editor of GMD, I would like to bring to your attention our Editorial version 1.1:

http://www.geosci-model-dev.net/8/3487/2015/gmd-8-3487-2015.html

This highlights some requirements of papers published in GMD, which is also available on the GMD website in the 'Manuscript Types' section:

http://www.geoscientific-model-development.net/submission/manuscript_types.html

[Figure]

In particular, please note that for your paper, the following requirement has not been met in the Discussions paper:

- "The main paper must give the model name and version number (or other unique identifier) in the title."

Therefore include the models name/acronym and its version number (MLUCM v1.0.0) in the title of your article in your revised submission to GMD.

Yours,

Astrid Kerkweg

---

## Referee Comment (RC2) · Anonymous Referee #2 · 19 Nov 2019

General comments

This paper presents the improvements of a 1D column model, previously developed on the basis of 3D RANS simulations, which is updated with new LES results. These LES results are obtained for several aligned and staggered arrays of cubes in a periodic domain, with varying plan area density, and with the LES model PALM.

The paper is already quite interesting and useful as it is but should benefit from some clarifications and improvements as described below. There is also a noteworthy sensitivity discussion of the results.

My first remark is on the use of "RANS" throughout the paper. RANS is an equation

system and need to be used with a turbulence closure. In the paper it appears that RANS is synonymous with RANS+ "k-l" turbulence closure. This is not trivial because a lot of the shortcomings mentioned are for the k-l closure and not RANS with more general closure. This should be clarified in the paper.

My second remark concerns the sentence in the introduction: "RANS simulations as the basis for 1-D parameterization. Given the simplified assumption of the turbulent flow in the RANS models, it is likely that the turbulent length scales derived from the RANS-CFD model are a culprit". This sentence is very paradoxical because the 1D model is also RANS-k-l. It would be nice to explain why we expect RANS to perform better in 1D than in 3D ? Another point might be that other levels of closure might not have these shortcomings. It is well known that the k-l closure is not good for obstacles and the k-eps approach is much better for that. Why not choose such a closure or even a second order closure (it is cheap in 1D) ?

My third remark is regarding the analysis of the length scale (lk). It is presented p14 as a "measure of the efficiency of vertical transport" while we see in Fig 9 that is has a minimum at roof level. However it is well known that there is a large transport at roof level (Fig.12 right), caused by large instationary structures induced by the roofs. This is very paradoxical and can work only in conjunction with a very sharp velocity gradient (equation 2 and 3). Here it seems to me that the model is trying to compensate for a too large velocity gradient with a small lk. Another turbulence closure might be able to overcome this.

Another new aspect of this paper is to include the dispersive stress in the 1D parameterization. While a good case is made in Fig 9.2 that the length scale variation as a function of plan area density is more realistic with LES results, it would be interesting to add the RANS results also with the dispersive stress included to see if this is still the case.

Finally there are some problems with the results with trying to cast the LES results

into a simple k-l closure and also maybe with the averaging used that need further discussion / explanation: - Fig 6 left : there seems to be negative values for the average canopy velocity profile - Fig. 9 left : the length scale becomes negative - Fig. 10 right : for the Cmu constant first we have to swallow its variation with height and then it becomes negative !

Detailed comments :

P7L11 : "neutral simulation for idealized configuration" : how much of these results can we expect to hold with stratifications and irregular neighborhood?

P7L24 : "The flow is driven by a pressure gradient of . . .. The corresponding u_T is 0.2ms−1" How is this total wall friction velocity obtained ? What about the pressure drag on the obstacles : it is not mentioned in the paper (and could be computed)?

P8L8 : RANS k-l closure ?

P8L18 : for the choice of domain size : the laboratory studies are not periodic and therefore they must indicate a number of row necessary to reach nearly constant values independent of position. (maybe 4-6 rows ?). Can you compare your results to these laboratory studies ?

P9L9 : spin up time of 3h seems enormous for such a small area. The rest of the paper is adimensional so it is difficult to judge (grid/domain size, velocity ?) but the time step of 2s seems also large. Why is there a need of a sampling every 50 time step ? Averages could be computed along the calculation if it is a storage problem.

P11L8 : in the discussion of the dispersive stress (Fig 7) what is the significance of the change of sign. How can it be interpreted?

P15L4 different zones is : are

P17L31 : the same diffusion coefficient is due to the k-l chosen but is different in the k-eps

---

## Author Comment (AC1) · 12 Jan 2020

Dear Dr. Astrid Kerkweg,

Thank you for sharing your remarks and the reviewers' comments on our manuscript GMD-2019-230. We appreciate this opportunity to revise and improve the manuscript and addressed all comments raised by the reviewers in the following document.

With regards to the editorial remarks and requirement for publication at GMD, we note that the name of the model is now included in the manuscript title: *"A One-Dimensional Model of Turbulent Flow Through 'Urban' Canopies (MLUCM v2.0): Updates Based*

[Figure]

*on Large-Eddy Simulation".* The present study is noted as version 2.0 to acknowledge the previous urban canopy parameterization developed by Santiago and Martilli 2010 using RANS results.

We believe that the modifications based on these comments have improved the quality of the manuscript and hope that this revised manuscript will meet your expectations.

Best,
Negin Nazarian
On behalf of all co-authors

[Figure]

**1 Review Comments by Referee #1:**

**General considerations**:
The paper is sound in its analysis (sometimes a little too optimistic in its conclusions, though) and the experimental plan, but not very carefully written (see the long list of minor comments). Also, the language could be improved (I have only indicated the obvious language errors). I have got two 'rather major' comments, of which the first is easily addressed, and the second may help to extend the speculations on why even the LES-based parameterization fails to reproduce some of the characteristics in real building urban configurations. Once the comments are properly addressed, the paper can certainly be recommended for publication in GMD.

**Response:** We thank the reviewer for the detailed review of this manuscript and are pleased that the contribution is deemed valuable for publication. The comments raised and proposed modifications were valid and helped us improve the quality of writing and explanations. We have addressed all remarks as detailed below and hope that the revised manuscript meets the standards for GMD publication.

**Major comments**:

1. The paper is based on an earlier finding of the authors (or some of them) that the BEP-tree underestimates the vertical exchange of momentum and energy in the canopy (p3, l.25). This is essentially concluded from comparison to one experiment in Vancouver. I trust that the authors have also assembled other evidence from earlier validations of the model. It would be more convincing to support the motivation for the paper to briefly summarize this additional evidence – or at least make a reference to where it can be found.

   **Response:** Thank you for this remark. Although this work was initially motivated by the analyses done by BEP-tree, the underestimation of turbulent fluxes is confirmed and shown directly in this paper using the comparison of LES and RANS

results (Fig. 5). This finding, in fact, is not new: The RANS model by Santiago and Martilli (2010) was previously compared extensively against the wind-tunnel experiments of Brown et al. 2001 (detailed in Santiago et al. 2007) and substantial underestimation of TKE was seen within the canopy as well as the area right above the building height ($z/H < 1.5$). In our comparison with Brown et al. 2001, we observe that LES results show substantial improvement compared to the RANS model in the prediction of the TKE profile within the canopy. Additionally, when the updated parameterization is used in the BEP-Tree model, we find improved performance not only compared to Vancouver measurements, but against measurements in London and Salt Lake City. These comparisons are detailed in the forthcoming publication by the authors (Krayenhoff et al., 2020). This information is now included in the Summary and Conclusions section of the manuscript:

"*Additionally, when the updated parameterizations were used in the BEP-Tree model (Krayenhoff, 2014), we observed improved performance compared to measurements taken across the diurnal cycle at three sites located in Vancouver (BC) and London (ON) in Canada, and Salt Lake City (UT) in USA (Krayenhoff et al., 2020).*"

2. P5, l.14 (u and w. . .): by neglecting the lateral component (and the corresponding covariances) the authors only consider frictional stress and not directional stress (v'w' – including its dispersive contribution). At the top of the canopy – and depending of course on the wind direction relative to the orientation of the canyon axis – directional stress might be quite important as well. Can the authors comment on this?

**Response:** We agree with the reviewer that the lateral components are not considered in the parameterization analyses presented here. This is due to the assumption that wind speed is orthogonal to obstacle faces (the building facade), which results in zero directional stress and lift. However, we note that the analyses of wind direction on the parameterization of the drag coefficient has been addressed by one of the co-authors in Santiago et al. (2013) and it was concluded that a height-dependent drag coefficient is needed to capture the lateral effects within the canopy for oblique wind directions. Indeed, for real building configurations and wind directions, the lateral components will play a role. Ideally, to account for the street and wind directions in realistic configurations, we need a methodology to derive dominant street directions over each grid cell and compute the drag coefficient as a function of height and the angle between street and wind direction above the canopy. This is particularly complex and requires a deep understanding of physical phenomena at the street canyon, which motivates starting the present analyses with simplified configurations. Future work needs to build upon work done by Santiago et al. (2013) to develop such methodologies and assess realistic configurations and wind direction conditions. This is now noted in the Summary and Conclusions section in the manuscript:

"*Further analysis is also needed to fully evaluate the effects of idealized configurations in parameterizations and assess the impact of variable building heights and wind directions on turbulent length scales and drag parameterization. Santiago et al. (2013) showed that a height-dependent drag coefficient is needed to capture the lateral effects within the canopy for oblique wind directions. To further account for the street and wind directions in realistic configurations, future work is needed to develop a methodology that derives dominant street directions over each grid cell and computes the drag coefficient as a function of height and the angle between street and wind direction above the canopy.*"

**Minor comments**

1. Eq 8 I think, the ending | of the 'absolute value' ($< u >$) is missing here. Also, the rhs has dimensions of ms-2 (which corresponds to the usual acceleration due to

the pressure gradient force), but the lhs is from eq. (1), which is multiplied with the density – so the rhs needs to be multiplied by density as well.

**Response:** The | sign, as well as the density fraction, have been missing here which made this equation incorrect. We thank the reviewer for pointing this out and have corrected the equation.

2. P7, l.18 uniform grid resolution: but what it is? Also 0.03H in the horizontal?
**Response:** The uniform grid resolution is the same in x and y direction, as well as $z$ direction up to $4H$. This is now clarified in the text:

"*In all simulations, the canyon height is resolved by 32 grids and the same uniform grid resolution is used in $x$ and $y$ directions ($0.0312H$). In the vertical direction ($z$), a uniform grid resolution is used up to $4H$ and grid spacing is gradually increased thereafter.*"

3. P8, l.5 Fig. 4: is this the aligned or the staggered configuration from which the modeled data is averaged?
P8, l. 17 7.4H is used to ensure ... so, is Fig. 4 presenting results with the 7.4H domain height? Same (below) for the number of obstacles in the domain. Both should be mentioned in the caption of Fig. 4.
P8, l.17 what is 'the lack of solution [for the entire BL]'?
Fig. 4 legend: what are the letters 'M', 'O' and 'Q' referring to? Also, in the text, k is used for TKE – so, it should be so in the figure.
**Response:** We acknowledge that important information regarding this comparison has been missing in the submitted manuscript. We agree with the reviewer that such information is critical and have added the following to the text as well as the graph caption:
*"we compared the TKE profiles obtained with the LES results with the wind-tunnel experiment of Brown et al. (2001) for a 3D building array with aligned configurations and observed good agreement in the shape of the profiles and TKE above*

*the canyon, while an underestimation of TKE within the building levels is seen."*
Figure 4 caption: *"Comparison of the TKE profile at the center of the canyon with experimental results of Brown et al. (2001) for a 3D building array with aligned configurations ($11 \times 7$ obstacles). The aspect ratio of the wind-tunnel experiment and numerical simulations are set to one ($H/W = 1$), resulting in the skimming flow regime (Oke, 2002). The domain height in the numerical simulations was set to $8H$ to be compatible with experimental set up as well as numerical results of Santiago et al. (2007). Vertical profiles along the centerline of the last three street canyons (indicated by M, O, Q here) are compared with the ensemble-averaged vertical profile in the LES simulations. More information regarding the experiment configuration and comparison with numerical results can be found in Brown et al. (2001) and Santiago et al. (2007)."*

4. P10, l. 17 a more accurate flow model: but still, Fig. 4 suggests that PALM underestimates TKE (by a factor of 2 or so in the mid- canyon). Can the authors comment on this?
**Response:** We agree that LES results also underestimate the TKE value within the canopy and note that such underestimation in the canopy compared to measurements was also reported in other LES results such as Giometto et al. (2016). However, in Figure 5, we show that compared to LES, RANS model results in an even lower value for TKE, confirmed by the previous comparison of RANS results with Brown et al. 2001 experiment (Santiago et al., 2007). This, as well as other factors that may contribute to underestimation of TKE within the canopy is further elaborated in the text:

*"Such underestimation of TKE compared to measurements in the canopy was also reported in previous studies such as Giometto et al. (2016) for a realistic urban configuration. Additionally, since the exact value of friction velocity was not available in the experimental dataset, the velocity at $3H$ is used for this comparison which may further contribute to the discrepancy. A direct comparison*

[Figure]

*between LES and RANS demonstrates that RANS underestimates TKE even further compared to the wind tunnel results (Sect. 2.2.3)."*

5. Figs. 5 and 6 the two figures share the same information for the three cases of Fig. 5 but they have different colors for the same configuration (what is yellow/brownish dots in Fig. 5 is a green dashed line in Fig. 6. It would be very helpful if these colors were the same.
**Response:** Thanks for the suggestion. We have made the colors and line styles in Fig. 5 compatible with Fig. 6.

6. Fig. 7, caption 'a)' and 'b)': the panels do not have the labels a) and b). Also, it is a little disturbing, that the two panels do not have (exactly) the same vertical axis.
**Response:** The caption is now corrected and the vertical axes in plots are made compatible:

*"Left: Vertical (spatially and temporally averaged) profiles of normalized dispersive stress $\langle \tilde{u}\tilde{w} \rangle / u_\tau^2$, and right: the contribution of dispersive stress to the total turbulent momentum flux $\langle \tilde{u}\tilde{w} \rangle / (\langle \overline{u'w'} \rangle + \langle \tilde{u}\tilde{w} \rangle)$. "A" in this graph indicates "Aligned" configuration, while "S" stands for "Staggered". "*

7. P14, l.13 are in good agreement: first, the authors probably want to refer to Fig. 8 (which is never mentioned in the text). Second, this figure shows that the 'good' agreement is rather qualitative than quantitative for small lambdas (e.g. for lambda=0.11, the difference in $C_deq$ is almost a factor of two)
**Response:** We note that Fig. 8 is noted in the text in Pg15, l.3. Nonetheless, we agree that the description here should be updated to clarify that LES results are higher than RANS:

*"Following this method, the drag coefficient parameterization using the LES results is shown in Fig. 8 ... Comparing the LES and RANS results, the trends in*

*$Cd_{eq}$ with $\lambda_p$ are in good agreement, but as previously demonstrated by Simón-Moral et al. (2014), RANS tend to overestimate the value of $C_{deq}$."*

8. the notation: $C_{deq}$ is what? The sectorial drag coefficient or an 'equilibrium' drag coefficient? Also, the notation 'DELTA  || for the horizontally averaged mean pressure deficit seems to be wrong. It should rather be something like <DELTAp(z)> (with an overbar on p)

    **Response:** The definition of $C_{deq}$ is explained in Section 3.2 of the manuscript based on the sectional drag coefficient proposed in Santiago and Martilli (2010). Unfortunately, it is not clear to authors what the reviewer is referring to by noting mean pressure deficit as wrong but we hope that the description here as well as in Santiago and Martilli (2010) (Section 5) is sufficient to clarify.

    *"It is known that the sectional drag coefficient depends on the packing density and the configuration of the array with a strong dependency with height, such that $C_d = C_d(z, \lambda_p)$ (Macdonald, 2000; Santiago et al., 2008; Santiago and Martilli, 2010). However, as indicated by Santiago and Martilli (2010), height-dependent parameterization of drag coefficients is challenging due to the high variability of $C_d$ close to the ground due to small $\langle \overline{u} \rangle$ as well as the lack of experimental information on the vertical profiles of this property inside the urban canopy. Accordingly, Santiago and Martilli (2010) proposed the following calculation of equivalent drag coefficient that is constant with height in the urban canyon, considering that "when integrated in the whole urban canopy, the drag force must be equal to that computed by the CFD simulations". "*

$$C_{deq} = \frac{\frac{-1}{\overline{\rho} H} \int_0^H \Delta \langle \overline{p(z)} \rangle dz}{\frac{1}{H} \int_0^H \langle \overline{u(z)} \rangle | \langle \overline{u(z)} \rangle | dz}$$

9. Eq 11 in this equation, and together with the text, the length scale $l_{kM}$ is introduced without explicitly mentioning it. In eq (7), this was $l_k$ (as it is in the text), and in (11a) it is $l_t$. All this must be carefully introduced, so that the reader knows

what is 'generic' (I assume $l_k$) and what is particular.

**Response:** We agree with this remark and have modified the text and graphs to clearly distinguish between $l_k$ and $l_{kM}$. In this format, redefining $l_{kt}$ was not necessary and is removed in the revised manuscript.

*"Combining Eqs. 3 and 4, the turbulent length scale $C_k l_k$ is traditionally calculated only considering the Reynolds stress $\langle u'w' \rangle$ (Eq. 11a). Here, following the discussions in Sect. 3.1, we recalculate turbulent length scale using total momentum fluxes that include turbulent dispersive flux $\langle \tilde{u}\tilde{w} \rangle$, shown as $C_k l_{kM}$ in Eq. 11b."*

10. Fig.9 legend refers to eqs. 11 and 12, but should be 11a and 11b.

    **Response:** Thank you for pointing out this mistake. The figure legend is now corrected.

11. P14, l.33 h-d: I assume 'd' is the zeroplane displacement height, but this has not been introduced so far. And again, canopy height is now 'h' rather than 'H' as previously.

    **Response:** We apologize for such inconsistencies in the manuscript. This is now corrected. The definition of "d" as zero-plane displacement height is also included in the manuscript.

    *"... inside the canopy, the length scale is mostly constant with height (specifically for $\lambda_p \geq 0.25$) as it is controlled by the shear layer ($H - d$, where $d$ is the zero-plane displacement height) ... "*

12. P15, l.5 $L_{eps}$ is now capitalized (also in Fig. 9 and its caption), while it was '$l_{eps}$' in eqs (6) and (7) and on p15, l.1 (and last paragraph of p14). All this must be consistent throughout the paper. Response: We acknowledge this discrepancy and have corrected this term as "$l_{eps}$" throughout the manuscript. Fig change

    **Response:** We acknowledge the discrepancy. "$l_\varepsilon$" is now consistently used

throughout the manuscript and graphs.

13. Fig. 11 what are the 'double overbars' referring to here? (eq. (15), for example, does not have it.)
    **Response:** This indicator is now defined in the figure caption:
    *"The results are obtained using the staggered urban configuration (Fig. 11) and averaged in the canopy volume ($\overline{\overline{Q}}$ here indicates volume-average of quantity $Q$)."*

14. P17, l.27 especially at the canopy level: if you mean with this, z/H=1 exactly, this is probably not true (at z/H=1, for all three densities, the red line is closer to the blue circles. So, where exactly do you mean?)
    **Response:** We agree with the reviewer that this was not clearly described so the text is now corrected as follows:
    *"Overall, the prediction of the horizontally-averaged velocity is improved compared to 1D-RANS, particularly within the canopy ($z/H < 1$) ... "*

15. P17, l.32 what is the 'turbulent equation'?
    **Response:** This is now corrected to "TKE" equation.

16. Fig.14, caption in the legend, new abbreviations are introduced ('G2017', 'N2019' etc. which need to be explained in the caption).
    **Response:** We agree with this comment and have updated the figure and caption. Please note that N2019 is removed as it refers to the present study.

    *"Comparison of vertical profile of velocity ($\langle \overline{u} \rangle / u_\tau$), turbulent kinetic energy ($\langle \overline{k} \rangle / u_\tau^2$), and turbulent momentum flux ($\langle \overline{u'w'} \rangle / u_\tau^2$) obtained with multi-layer (1-D) model with the (3-D) LES results of Giometto et al. (2017) for realistic urban configurations (G2017) as well as LES simulations discussed here for idealized configurations."*

**Editorial Comments:**

- P2, l.7. from the street-scale to synoptic scales.

- P2, l.13 'is being treated' may be better

- P3, l32 I'd use 'hypothesize' rather than speculate

- P6, l. 13 is calculated..

- P7, l. 5 the simulation domain

- P7, l. 17 the canyon height

- P8, l. 9 compared to the wind tunnel results: it has not yet been mentioned that the Brown et al. study has used wind tunnel experiments.

- Eq (10) one is inclined to assume that 'h' in the integrals corresponds to the canopy height - but this has been denoted 'H' before. This must be consistent.

- Fig. 5 caption: 'compared with'. WHAT is compared to RANS? (I assume the PALM output but needs to be mentioned)

- P11, l 7 the total flux is given as <overbar(u'w')> + <overbar(u'w')>. One of the two terms should be the dispersive flux. (same in the caption of Fig. 7)

- P14, l.3 are discussed

- P14, l. 32 the length scale: : :

- P15, l.1 correspond to

- P15, l. 4 are then defined

- P17, l.21 Fig. 12 shows (does not demonstrate)

- P19, l.11 result in a substantial

- P19, l.16 results in relatively

- P20, l.2 parameterizations

- P20, l9 of the spatially: : :

- P20, l.14 dissipation length scale

- P21, l.2 assumption that the diffusion: :
  **Response:** Thank you for providing such a detailed review of our manuscript. All these remarks are now addressed.

**2 Review Comments by Referee #2:**

The paper is already quite interesting and useful as it is but should benefit from some clarifications and improvements as described below. There is also a noteworthy sensitivity discussion of the results.

**Response:** We thanks the referee for the review of this paper and are pleased that the analyses are deemed interesting. Comments raised here are addressed in the following document and we hope the updated manuscript meets the standards for GMD publication.

1. My first remark is on the use of "RANS" throughout the paper. RANS is an equation system and need to be used with a turbulence closure. In the paper it appears that RANS is synonymous with RANS+ "k-l" turbulence closure. This is not trivial because a lot of the shortcomings mentioned are for the k-l closure and not RANS with more general closure. This should be clarified in the paper:

   **Response:** Thank you for pointing out the missing information regarding the RANS model that was the base of the previous multi-layer model. The CFD simulations in Santiago and Martilli 2010 are based on the steady-state RANS with the standard $k - \varepsilon$ turbulence model that solves two transport equations, one for turbulent kinetic energy and one for the dissipation. This is now clarified in the manuscript when referring to the previous model.

   *"This model employs horizontally-averaged microscale Computational Fluid Dynamics (CFD) simulations based on Reynolds-Averaged Navier-Stokes (RANS) with the standard k–ε turbulence model to determine required input parameters to the column model "*

2. My second remark concerns the sentence in the introduction: "RANS simulations as the basis for 1-D parameterization. Given the simplified assumption of the turbulent flow in the RANS models, it is likely that the turbulent length scales derived

from the RANS-CFD model are a culprit". This sentence is very paradoxical because the 1D model is also RANS-k-l. It would be nice to explain why we expect RANS to perform better in 1D than in 3D ? Another point might be that other levels of closure might not have these shortcomings. It is well known that the k-l closure is not good for obstacles and the k-eps approach is much better for that. Why not choose such a closure or even a second-order closure (it is cheap in 1D) ?

**Response:** Thank you for your remark. We agree with the reviewer that RANS with the $k - \varepsilon$ closure model is more suitable for flow simulations in building array configurations. The RANS model used in Santiago and Martilli did, in fact, use the $k - \varepsilon$ closure. Indeed, we do not expect the 1D parameterization in RANS to outperform the 3D results. However, in order to obtain reasonable parameterizations that can accurately predict the horizontally-averaged flow within the canopy, it is important that the underlying 3D results are of high accuracy and able to resolve flow characteristics relevant to spatially-averaged flow. Regardless of $k - \varepsilon$ or $k - l$ closure model, it is shown that RANS models fall short in accurately predicting the TKE distribution within and above the canopy, which is our motivation to move to LES modeling. Regarding the use of the $k - l$ closure model, we note that the overarching goal of the 1-D model is to be implemented in mesoscale climate modeling. In mesoscale models, $k - l$ approximations are far more common given the good performance of k-theory for wall-flow, as well as being a computationally efficient procedure that directly accounts for the length scales from parameterizations. The use of $k - \varepsilon$ closure is indeed an interesting alternative approach that can be addressed in future research, however, given that a) there is no evidence that $k - \varepsilon$ closure performs better in 1-D, b) there is a need for computationally efficient subgrid-scale calculation in mesoscale simulations, and c) extensive work is done on the $k - l$ multi-layer model over the past two decades, we believe that the choice of $k - l$ closure model is reasonable here.

3. My third remark is regarding the analysis of the length scale (lk). It is presented p14 as a "measure of the efficiency of vertical transport" while we see in Fig 9 that is has a minimum at roof level. However it is well known that there is a large transport at roof level (Fig.12 right), caused by large instationary structures induced by the roofs. This is very paradoxical and can work only in conjunction with a very sharp velocity gradient (equation 2 and 3). Here it seems to me that the model is trying to compensate for a too large velocity gradient with a small lk. Another turbulence closure might be to overcome this.

**Response:** Thank you for this remark. We first note that the minimum turbulent length scale for all $\lambda_p$ cases is at the ground level where the Reynolds stress is zero. Above ground level, $l_k$ increases with height in the canopy (as the velocity gradient decreases mid-canyon) and decreases afterward as it gets closer to the roof level (as a result of large shear stress and TKE). Although buildings cause turbulent structures at their scale, the roof also causes small turbulent structures just above roof height (due to the small mixing lengths close to that surface). The average of both of these effects likely produces a length scale minimum.

Here, it is important to note that the turbulent length scale is directly derived from the LES results and expresses the relative contribution of three terms in the canopy: a) Reynolds Stress, b) TKE, and c) shear stress. Nonetheless, the higher $l_k$ values indicate higher vertical transport of momentum (including turbulent and dispersive) for the same TKE and vertical flow gradient, which is referred to as the measure of the efficiency of vertical transport here. We do not believe this is modified by another choice of closure model particularly as the 3 terms are obtained from the resolved-scale LES results, which are assumed as the reference. The very sharp gradient at the roof level is not an artifact of the k-L model, but it is rather produced by LES.

4. Another new aspect of this paper is to include the dispersive stress in the 1D parameterization. While a good case is made in Fig 9.2 that the length scale

variation as a function of plan area density is more realistic with LES results, it would be interesting to add the RANS results also with the dispersive stress included to see if this is still the case.

**Response:** We agree that this comparison would have been interesting but as Santiago and Martilli 2010 neglected the inclusion of dispersive stress in their analyses, this data is not available to us. Nonetheless, here we aim to show that not only the choice of LES vs RANS is important, but also dispersive stresses should be included to fully represent the spatially-averaged flow within the urban canopy.

5. Finally there are some problems with the results with trying to cast the LES results into a simple k-l closure and also maybe with the averaging used that need further discussion / explanation: - Fig 6 left: there seems to be negative values for the average canopy velocity profile - Fig. 9 left : the length scale becomes negative - Fig. 10 right : for the Cmu constant first we have to swallow its variation with height and then it becomes negative !

**Response:** Indeed, there are certain simplifications in the k-l model that contribute to inaccuracies in the MLUCM model. A few of those are mentioned in the Conclusions section which will form the basis of our future analyses:

*"spatially-averaged turbulent kinetic energy, $\langle \overline{k} \rangle$ is still underestimated close to the ground for high $\lambda_p$ values due to the underestimation of turbulent transport deep in the canopy. Preliminary analyses of turbulent transport in this study (not shown) reveal that the K-theory assumption that the diffusion coefficient $K_m$ is the same for TKE and momentum equations (i.e. $K_m = \frac{\langle u'w' \rangle}{\partial \langle \overline{u} \rangle / \partial z} = \frac{\langle k'w' \rangle}{\partial \langle \overline{k} \rangle / \partial z}$) does not hold in the LES results. Accordingly, future work should revisit the multi-layer model formulations to assess 1) the parameterization of turbulent transport term in the 1-D TKE equation (Eq. 5) and 2) distinction between the diffusion length scale of momentum and TKE. " Nonetheless, despite the simplifications, MLUCM has been proven as a successful and efficient tool for representing sub-grid ur-*

*ban flow characteristics in meso-scale models. "*
Regarding the results noted by the referee: The negative value of velocity in the canyon for higher $\lambda_p$ cases has been demonstrated in the literature (particularly in the staggered configuration as noted by Herpin et al. (2018)). The negative value of Lk is only seen for $\lambda_p = 0.44$ for staggered configuration and is in fact not related to negative velocity values, but rather velocity gradient ($\frac{d<\overline{u}>}{dz}$) approaching zero mid-canyon and becoming negative at few grid points. This is then reflected in the calculation of $C_\mu$ as noted by the reviewer. We understand that this negative lengthscale and $C_\mu$ is neither physical nor feasible to represent in the 1-D model so in the final parameterization (Eq. 12-15 and Fig. 11), we excluded the negative values for $\lambda_p = 0.44$.

**Detailed comments**:

1. P7L11: "neutral simulation for idealized configuration": how much of these results can we expect to hold with stratifications and irregular neighborhood?
   **Response:** The answer highly depends on a) the ratio between thermal and momentum forcing in the canyon and b) urban configuration with respect to the prevailing wind direction. Regarding the impact of wind direction and urban configurations, we suggest reviewing Santiago et al. (2013) and Simón-Moral et al. (2014) that evaluated the impact of various wind angles and canyon spacing configurations on UCPs. Regarding the impact of thermal forcing, please review Santiago et al. (2014) that evaluated the variability in sectional drag based on a non-dimensional parameter representing the ratio between thermal and momentum forcing in the canopy. We also note that forthcoming Krayenhoff et al. (2020) have added temperature, humidity and buoyancy effects to the Krayenhoff et al. (2015) flow model and combined it with previously developed models on radiation (Krayenhoff et al., 2015) and thermal (Martilli et al., 2002) balance for a comprehensive representation of urban canyon and including trees at the street level.

This is further elaborated in the updated Summary and Conclusions section of the manuscript (in addition to the Introduction section previously noted):

"*Further analysis is also needed to fully evaluate the effects of idealized configurations in parameterizations and assess the impact of variable building heights and wind directions on turbulent length scales and drag parameterization. Santiago et al. (2013) showed that a height-dependent drag coefficient is needed to capture the lateral effects within the canopy for oblique wind directions. To further account for the street and wind directions in realistic configurations, future work is needed to develop a methodology that derives dominant street directions over each grid cell and computes the drag coefficient as a function of height and the angle between street and wind direction above the canopy. Lastly, the current study focused on the momentum exchange without considering the role of thermal forcing on turbulent length scales. Updated parameterization of thermal effects (investigated by Krayenhoff et al. 2020) can also be evaluated using LES results.*"

2. P7L24: "The flow is driven by a pressure gradient of... The corresponding $u_T$ is 0.2m/s" How is this total wall friction velocity obtained? What about the pressure drag on the obstacles: it is not mentioned in the paper (and could be computed)?
**Response:** The total wall friction velocity noted here can be computed using two methods: 1) from the prescribed magnitude of pressure gradient that is used to drive the flow (i.e. $\tau = \rho u_\tau^2 / H_t$, where $H_t$ is the total domain height and total shear stress $\tau$ is equivalent to the pressure gradient imposed in the horizontal direction), and 2) using the surface kinematic momentum fluxes in the horizontal directions (i.e. $u_\tau = \left( \overline{u'w'}^2 + \overline{v'w'}^2 \right)^{1/4}$). Both methods yeild the value of $0.2 m\,s^{-1}$ noted here. This is now added to the manuscript:
" *The flow is driven by a pressure gradient of magnitude $\tau = \rho u_\tau^2 / H_T$, where $u_\tau$ is the total wall friction velocity, and $H_T$ is the total domain height ($7.4H$). The*

*corresponding $u_\tau$ is $\approx 0.21\,\mathrm{m\,s^{-1}}$ which results in $Re_H = UH/\nu \approx 10^6$. We note that calculating $u_\tau$ using the surface kinematic momentum fluxes in the horizontal directions (i.e. $u_\tau = (\overline{u'w'}^2 + \overline{v'w'}^2)^{1/4}$) yields the same value.*"

3. P8L8: RANS k-l closure?
   **Response:** RANS model noted here uses the $k - \varepsilon$ closure model. This is now noted in the text.

4. P8L18: for the choice of domain size: the laboratory studies are not periodic and therefore they must indicate a number of rows necessary to reach nearly constant values independent of position. (maybe 4-6 rows ?). Can you compare your results to these laboratory studies?
   **Response:** We acknowledge that important information regarding the comparison with the experimental study has been missing in the submitted manuscript. Such information is critical here and are added to the text as well as the graph caption:
   *"we compared the TKE profiles obtained with the LES results with the wind-tunnel experiment of Brown et al. (2001) for a 3D building array with aligned configurations and observed good agreement in the shape of the profiles and TKE above the canyon, while an underestimation of TKE within the building levels is seen."*
   Figure 4 caption: *"Comparison of the TKE profile at the center of the canyon with experimental results of Brown et al. (2001) for a 3D building array with aligned configurations ($11 \times 7$ obstacles). The aspect ratio of the wind-tunnel experiment and numerical simulations are set to one ($H/W = 1$), resulting in the skimming flow regime (Oke, 2002). The domain height in the numerical simulations was set to $8H$ to be compatible with experimental set up as well as numerical results of Santiago et al. (2007). Vertical profiles along the centerline of the last three street canyons (indicated by M, O, Q here) are compared with the ensemble-averaged vertical profile in the LES simulations. More information regarding the experiment*

*configuration and comparison with numerical results can be found in* Brown et al. *(2001) and* Santiago et al. *(2007)."*

5. P9L9: spin-up time of 3h seems enormous for such a small area. The rest of the paper is adimensional so it is difficult to judge (grid/domain size, velocity ?) but the time step of 2s seems also large. Why is there a need of a sampling every 50 time step? Averages could be computed along with the calculation if it is a storage problem.
**Response:** The spin-up time interval in the LES models tends to be much larger than RANS and depends on not only the size of the domain, but also size of the grids with respect of canyon vortex size and timestep. The conventional method to determine the spin-up time in LES results is to monitor the temporal evolotion of the volume-average TKE in the computational domain and discard the time interval (spin-up period) where <k> does not represent the quasi-steady behavior. The choice of time step and sampling interval is also tested in this study using a series of sensitivity analyses. For the sampling interval, it is important to note that quasi-steady behaviors in LES results happen with various frequencies. If the time-interval of sampling is too large, it is possible to skip such behaviors in the results, while smaller sampling frequency may not be necessary.

6. P11L8: in the discussion of the dispersive stress (Fig 7) what is the significance of the change of sign. How can it be interpreted?
**Response:** Positive values of dispersive fluxes within the canopy, of similar magnitude to the turbulent stress, implies that the flux is countergradient, indicating downward transport of slow air. This is now noted in the text.

7. P15L4 different zones is: are
**Response:** Thank you for pointing this out. It is now corrected

8. P17L31: the same diffusion coefficient is due to the k-l chosen but is different in the k-eps.
   **Response:** Please refer to our response to Major Comment 2. With due respect, we disagree that the diffusion coefficient is dependant on the closure model used here and believe the future work proposed to assess the diffusion coefficient for momentum and TKE equations based on the LES results is valid.

[Figure]

**References**

Brown, M. J., Lawson, R. E., DeCroix, D. S., and Lee, R.: Comparison of centerline velocity measurements obtained around 2D and 3D building arrays in a wind tunnel, Int. Soc. Environ. Hydraulics, Tempe, AZ, 2001.

Giometto, M., Christen, A., Meneveau, C., Fang, J., Krafczyk, M., and Parlange, M.: Spatial characteristics of roughness sublayer mean flow and turbulence over a realistic urban surface, Boundary-Layer Meteorology, 160, 425–452, 2016.

Giometto, M., Christen, A., Egli, P., Schmid, M., Tooke, R., Coops, N., and Parlange, M.: Effects of trees on mean wind, turbulence and momentum exchange within and above a real urban environment, Advances in Water Resources, 106, 154–168, 2017.

Herpin, S., Perret, L., Mathis, R., Tanguy, C., and Lasserre, J.-J.: Investigation of the flow inside an urban canopy immersed into an atmospheric boundary layer using laser Doppler anemometry, Experiments in Fluids, 59, 80, 2018.

Krayenhoff, E., Santiago, J.-L., Martilli, A., Christen, A., and Oke, T.: Parametrization of drag and turbulence for urban neighbourhoods with trees, Boundary-Layer Meteorology, 156, 157–189, 2015.

Krayenhoff, E., Jiang, T., Christen, A., Martilli, A., Oke, T., Bailey, B., Nazarian, N., Voogt, J., Giometto, M., Stastny, A., and Crawford, B.: A multi-layer urban canopy meteorological model with trees (BEP-Tree): Street tree impacts on pedestrian-level climate, Manuscript under review, Urban Climate, 2020.

Krayenhoff, E. S.: A multi-layer urban canopy model for neighbourhoods with trees, Ph.D. thesis, University of British Columbia, 2014.

Macdonald, R.: Modelling the mean velocity profile in the urban canopy layer, Boundary-Layer Meteorology, 97, 25–45, 2000.

Martilli, A., Clappier, A., and Rotach, M. W.: An urban surface exchange parameterisation for mesoscale models, Boundary-Layer Meteorology, 104, 261–304, 2002.

Oke, T. R.: Boundary layer climates, Routledge, 2002.

Santiago, J. and Martilli, A.: A dynamic urban canopy parameterization for mesoscale models based on computational fluid dynamics Reynolds-averaged Navier–Stokes microscale simulations, Boundary-layer meteorology, 137, 417–439, 2010.

Santiago, J., Coceal, O., Martilli, A., and Belcher, S.: Variation of the sectional drag coefficient of a group of buildings with packing density, Boundary-layer meteorology, 128, 445–457,

2008.

Santiago, J., Krayenhoff, E., and Martilli, A.: Flow simulations for simplified urban configurations with microscale distributions of surface thermal forcing, Urban Climate, 9, 115–133, 2014.

Santiago, J. L., Martilli, A., and Martín, F.: CFD simulation of airflow over a regular array of cubes. Part I: Three-dimensional simulation of the flow and validation with wind-tunnel measurements, Boundary-layer meteorology, 122, 609–634, 2007.

Santiago, J. L., Coceal, O., and Martilli, A.: How to parametrize urban-canopy drag to reproduce wind-direction effects within the canopy, Boundary-layer meteorology, 149, 43–63, 2013.

Simón-Moral, A., Santiago, J. L., Krayenhoff, E. S., and Martilli, A.: Streamwise versus spanwise spacing of obstacle arrays: parametrization of the effects on drag and turbulence, Boundary-layer meteorology, 151, 579–596, 2014.